# STAT3β Enhances Sensitivity to Concurrent Chemoradiotherapy by Inducing Cellular Necroptosis in Esophageal Squamous Cell Carcinoma

**DOI:** 10.3390/cancers13040901

**Published:** 2021-02-21

**Authors:** Zhen-Yuan Zheng, Ping-Lian Yang, Wei Luo, Shuai-Xia Yu, Hong-Yao Xu, Ying Huang, Rong-Yao Li, Yang Chen, Xiu-E Xu, Lian-Di Liao, Shao-Hong Wang, He-Cheng Huang, En-Min Li, Li-Yan Xu

**Affiliations:** 1The Key Laboratory of Molecular Biology for High Cancer Incidence Coastal Chaoshan Area, Shantou University Medical College, Shantou 515041, China; 18zyzheng1@stu.edu.cn (Z.-Y.Z.); yang_pinglian@126.com (P.-L.Y.); lavoury@163.com (W.L.); yushuaixia@163.com (S.-X.Y.); 19ryli@stu.edu.cn (R.-Y.L.); 16ychen13@stu.edu.cn (Y.C.); 2Department of Biochemistry and Molecular Biology, Shantou University Medical College, Shantou 515041, China; 3Institute of Oncologic Pathology, Shantou University Medical College, Shantou 515041, China; xexu@stu.edu.cn (X.-E.X.); ldliao@stu.edu.cn (L.-D.L.); 4Departments of Radiation Oncology, Shantou Central Hospital, Affiliated Shantou Hospital of Sun Yat-sen University, Shantou 515041, China; xinlingjifeng@163.com (H.-Y.X.); huanghecheng7373@163.com (H.-C.H.); 5Departments of Pathology, Shantou Central Hospital, Affiliated Shantou Hospital of Sun Yat-sen University, Shantou 515041, China; 13414034690@163.com (Y.H.); w196303@sohu.com (S.-H.W.)

**Keywords:** STAT3β, concurrent chemoradiotherapy, necroptosis, esophageal squamous cell carcinoma

## Abstract

**Simple Summary:**

The prognosis of esophageal squamous cell carcinoma (ESCC) patients is poor, with a five-year survival of 15–34%. We examined the expression of STAT3α and STAT3β in pretreatment tumor biopsies of 105 ESCC patients who received concurrent chemoradiotherapy (CCRT) by immunohistochemistry. The data showed that ESCC patients who demonstrate both high STAT3α expression and high STAT3β expression in the cytoplasm have a significantly better survival rate. Moreover, the ESCC patients with high STAT3β expression have a complete response to concurrent chemoradiotherapy. STAT3β-overexpressed ESCC cell lines exhibit CCRT (platinum plus radiation therapy) sensitivity, resulting in cell death. RNA sequencing found that ESCC cells highly expressing STAT3β undergo necrosis after CCRT. In summary, STAT3β could be potentially used to predict the response to CCRT, which may provide an important insight into the treatment of ESCC.

**Abstract:**

Concurrent chemoradiotherapy (CCRT), especially platinum plus radiotherapy, is considered to be one of the most promising treatment modalities for patients with advanced esophageal cancer. STAT3β regulates specific target genes and inhibits the process of tumorigenesis and development. It is also a good prognostic marker and a potential marker for response to adjuvant chemoradiotherapy (ACRT). We aimed to investigate the relationship between STAT3β and CCRT. We examined the expression of STAT3α and STAT3β in pretreatment tumor biopsies of 105 ESCC patients who received CCRT by immunohistochemistry. The data showed that ESCC patients who demonstrate both high STAT3α expression and high STAT3β expression in the cytoplasm have a significantly better survival rate, and STAT3β expression is an independent protective factor (HR = 0.424, *p* = 0.003). Meanwhile, ESCC patients with high STAT3β expression demonstrated a complete response to CCRT in 65 patients who received platinum plus radiation therapy (*p* = 0.014). In ESCC cells, high STAT3β expression significantly inhibits the ability of colony formation and cell proliferation, suggesting that STAT3β enhances sensitivity to CCRT (platinum plus radiation therapy). Mechanistically, through RNA-seq analysis, we found that the TNF signaling pathway and necrotic cell death pathway were significantly upregulated in highly expressed STAT3β cells after CCRT treatment. Overall, our study highlights that STAT3β could potentially be used to predict the response to platinum plus radiation therapy, which may provide an important insight into the treatment of ESCC.

## 1. Introduction

Signal transducer and activator of transcription 3 (STAT3) was initially identified as an acute-phase response factor activated by cytokines and is also activated by growth factors and oncogenes [1,2,3]. Once STAT3 is phosphorylated on tyrosine 705, facilitating STAT3 dimerization, nuclear translocation and the initiation of transcription occur [4,5]. STAT3 is aberrantly hyperactivated in various human cancers and correlates with a poor prognosis. It is crucial for regulating cell proliferation, resistance to apoptosis, angiogenesis, invasion, and metastasis [2,6,7,8]. In addition, STAT3 is associated with senescence, and blockage of STAT3 induces cellular senescence, which confers ionizing radiation resistance [9]. Accumulating evidence shows that STAT3 carries tumor suppressor functions and the functions of STAT3 are dependent on co-existing biochemical defects or genetic background [10,11,12]. Clinical observations also support the tumor suppressor role of STAT3 in head and neck cancer and breast cancer [13,14,15,16]. For instance, STAT3 knockout mice demonstrated higher levels of astrocyte proliferation and invasion [10]. Furthermore, conditional ablation of STAT3 in intestinal epithelial cells was shown to enhance the progression of benign adenomas in Apc^Min^ mice [11]. 

Despite *STAT3* being most commonly presented as an oncogene, its opposing role as a tumor suppressor also depends on its different isoforms [17,18,19,20], i.e., full-length STAT3α and truncated STAT3β, which are generated by alternative splicing of exon 23. As compared with STAT3α, STAT3β lacks the C-terminal transcriptional domain, but has an identical amino acid sequence, excepting the 55 amino acid residues at the C-terminal of STAT3α, which are replaced by seven unique amino acid residues [17,20]. STAT3β has been found to inhibit the transcriptional ability of STAT3α and act as a significant transcriptional regulator of its target gene expression [21,22,23]. Several studies have shown that STAT3β plays a tumor suppressor role in melanoma, breast cancer, esophageal carcinoma, lung cancer, acute myeloid leukemia, and gastric cancer [24,25,26,27,28,29,30]. For instance, STAT3β-transfected gastric cancer cells inhibit cellular migration and reduce chemoresistance [30].

Necroptosis is a programmed cell death induced by a variety of stimuli, such as tumor necrosis factor receptor (TNFR) superfamily receptors, pathogen infections, and Toll-like receptors [31]. The core components of the necroptosis pathway are receptor-interacting serine/threonine-protein kinases 1 (RIPK1), RIPK3, and the pseudokinase, mixed lineage kinase domain-like protein (MLKL) [32]. Escape and resistance from programmed cell death are considered to be essential hallmarks of cancer [33]. Therefore, necroptosis is a promising target for cancer therapies. However, previous studies reported that the expression of necroptotic pathway key regulators is downregulated in cancer cells [32,34,35], suggesting that necroptosis plays a curious wider role in tumor growth and cell death.

Esophageal carcinoma is one of the most aggressive cancers, ranking sixth in terms of incidence and seventh in terms of causing of cell death worldwide [36]. Esophageal squamous cell carcinoma (ESCC) is the major histological subtype in Asia and concurrent chemoradiotherapy (CCRT) is one of the most promising advanced ESCC treatments [37,38]. Even ESCC patients have diverse responses to CCRT, thus sensitive and specific molecular biomarkers for predicting CCRT response is an urgent problem. Enforcing STAT3β expression decreased the clonogenic ability of ESCC and enhanced the sensitivity to cisplatin and 5-fluorouracil [27]. However, the underlying mechanism of STAT3β to CCRT in ESCC remains unclear. In the current investigation, we show that STAT3β increases sensitivity to CCRT (platinum plus radiation therapy). Our data confirm that STAT3β induces cellular necroptosis upon exposure to platinum plus radiation therapy via activating the TNF signaling pathway and transcriptionally activating cell necrosis-related genes such as RIPK1 and MLKL. These results suggest STAT3β to be a new marker for CCRT treatment in patients with esophageal cancer.

## 2. Results

### 2.1. Opposing Prognostic Significance of STAT3α and STAT3β in ESCC Patients with CCRT

Constitutive activation of STAT3 has been shown in various types of malignancies, and the expression of p-STAT3 has been recognized as a predictor of poor survival [39,40,41]. Our previous study also showed that moderate/strong STAT3β significantly prolongs both overall survival and recurrence-free survival in patients who received adjuvant chemoradiotherapy (ACRT), but the model of STAT3α and STAT3β in terms of prognosis is incomplete [27]. We collected pretreatment tumor biopsies from 105 ESCC patients who received CCRT and explored the correlation between STAT3α and STAT3β and overall survival. Notably, STAT3α is expressed in both the cytoplasm and nucleus (Figure 1A), while STAT3β is mainly expressed in the cytoplasm of tissues (Figure 1B). Kaplan–Meier analysis showed that longer overall survival depended on the localization of STAT3α. Low-expression of nuclear STAT3α was associated with longer overall survival (*p* = 0.036) (Figure 1C). However, high expression of cytoplasmic STAT3α correlated with longer overall survival without significant change (*p* = 0.065; Figure 1D). STAT3β has an opposite role in clinical prognosis as compared to STAT3α [27]. Our data highlighted that high-expression cytoplasmic STAT3β was significantly associated with longer overall survival (*p* = 0.005; Figure 1E). As shown in Table 1, both univariate and multivariate analyses using Cox regression showed that high-expression nuclear STAT3α was a risk factor for overall survival (Univariant, HR = 1.769, 95%confidence interval (CI) = 1.039–3.011, *p* = 0.036. Multivariant, HR = 1.937, 95% CI = 1.127–3.328, *p* = 0.017). Strikingly, high-expression cytoplasmic STAT3β was a protective factor for overall survival based on univariant analysis (HR = 0.458, 95% CI = 0.261–0.802, *p* = 0.006) and multivariant analysis (HR = 0.424, 95% CI = 0.241–0.748, *p* = 0.003). Moreover, we evaluated which model of STAT3β and STAT3α was correlated with a favorable prognosis. Combining cytoplasmic STAT3β and nuclear STAT3α for assessing the correlation with overall survival, high-expression cytoplasmic STAT3β and low-expression nuclear STAT3α is the best combination to indicate prolonged overall survival (*p =* 0.001, Figure 1F). Furthermore, high-expression cytoplasmic STAT3β and high-expression cytoplasmic STAT3α also indicated prolonged overall survival (*p =* 0.008, Figure 1G). Taken together, high-expression cytoplasmic STAT3β and low-expression nuclear STAT3α correlated with favorable clinical prognoses and prolonged survival in ESCC patients alone or in combination. 

### 2.2. High STAT3β Expression in ESCC Patients Is Associated with the Response to Platinum Plus Radiation Therapy

We found that high-expression STAT3β was moderately associated with ESCC patient’s response to CCRT (ratio of complete response (CR) = 31.4%, *p =* 0.043), especially with a platinum plus ionizing radiation therapeutic regime (ratio of CR = 40.6%, *p =* 0.014) (Figure 2A). The correlation between STAT3β and clinical parameters also demonstrated that high-express cytoplasmic STAT3β had a better response to CCRT (*p =* 0.008; Table 2). However, the STAT3α score and the analysis of the correlation between STAT3α and clinical parameters had no apparent impact on CCRT response (Figure 2B and Appendix A). These results suggested that ESCC patients with high expression levels of STAT3β who were receiving concurrent radiotherapy and chemotherapy tended to complete remission and had a favorable prognosis. 

### 2.3. STAT3β Overexpression Enhances Sensitivity to Chemoradiotherapy in ESCC Cells

To investigate the role of STAT3β in CCRT based on the clinical outcome, the expression of the STAT3 and STAT3 spliced isoform STAT3β was examined by western blotting in ESCC cell lines. As shown in Figure 3A, the ratio of STAT3β/STAT3α was very low in these seven ESCC cell lines. To further study the response to cisplatin (DDP), the IC50 of cisplatin was assessed the cisplatin-resistant ESCC cells. Regarding treatment with cisplatin for 24 h, KYSE150 was more resistant than the others with a half-maximal inhibitory concentration (IC50) of 44.6 μM, while TE3 was the most sensitive cells with IC50 5.49 μM (Figure 3B and Appendix A). Then, we detected the total and phosphorylated-Y705 of STAT3 by western blotting after treatment with cisplatin, ionizing radiation (IR), or a combination (cisplatin plus ionizing radiation, CCRT). We found that STAT3α Y705 phosphorylation declined in four cell lines after treatment with cisplatin or ionizing radiation, and this was more obvious with CCRT (Figure 3C). Furthermore, we analyzed the sensitivity to ionizing radiation and CCRT as compared to the relative resistance to cisplatin. As shown in Figure 3D, the pattern diagram exhibited the design of a colony formation assay. The results showed that radiotherapy combined with chemotherapy could inhibit tumor clonogenicity more effectively at certain concentrations and ionizing radiation doses (Figure 3E–H). 

We next examined whether highly expressed STAT3β could increase the sensitivity of ESCC cells to CCRT. We established stably expressing STAT3β ESCC cell lines, KYSE30 and KYSE150, which were labeled KYSE30-STAT3β and KYSE150- STAT3β (Figure 4A). Fluorescence staining was performed to determine the intracellular localization of STAT3α and STAT3β, when Flag-STAT3α and STAT3β-HA were transfected alone or combined in the KYSE150 STAT3-KO cell line. STAT3α and STAT3β were co-localization and distributed in nucleus and cytoplasm (Appendix A). We found that pSTAT3α was reduced even if it was relatively enhanced in the case of STAT3β overexpression (Figure 4B). We then investigated the effects of STAT3β overexpression on clonogenic survival under cisplatin, ionizing radiation, and CCRT treatment via clonogenic assay. High STAT3β expression significantly decreased the clonogenic ability compared with control cells when treated with different concentrations of cisplatin, various doses of ionizing radiation, or CCRT (Figure 4C,D and Appendix A). Thereafter, we further addressed the sensitivity to CCRT of highly expressed STAT3β cells. A cell proliferation assay was used for validation, in which the proliferation of cells was confirmed by EdU staining, both in the KYSE30, KYSE150-Vector, and -STAT3β groups. The EdU-positive cells were dramatically decreased after cisplatin, ionizing radiation, and CCRT treatment (Figure 4E,F). Thus, these data indicate that high STAT3β expression likely contributed to enhance the ESCC cells’ sensitivity to chemoradiotherapy. 

### 2.4. STAT3β Overexpression Enhances Cell Necroptosis after Chemoradiotherapy

The survival fraction suggested that STAT3β decreased the ability of tumor proliferation and increased sensitivity to CCRT. Next, we investigated the mechanism underlying STAT3β-mediated sensitization of CCRT. STAT3β has been determined as significant transcriptional regulator, regulating a distinct set of its own specific genes involved in cell organization, metabolism, and protein metabolism [23]. To further reveal the functions of STAT3β in CCRT ESCC cells, we conducted a transcriptome analysis to define its transcriptional roles. After analyzing these samples, a statistical significance cut-off was set at FDR < 0.05 and log FC of ≤–0.58 or ≥0.58 (a 1.5-fold change in differentially expressed genes (DEGs) in highly expressed STAT3β). Gene ontology (GO) biological process and Kyoto Encyclopedia of Genes and Genomes (KEGG) pathway analyses using the online Metascape tools revealed STAT3β upregulated genes involved in the TNF signaling pathway and necroptosis (Figure 5A,B). The TNF signaling pathway is the most significant up regulated in KEGG. Necroptosis is a form of programmed cell death, which is induced by ligand binding to TNF family death domain receptors, and necroptosis pathway also enrich in our RNA-seq results. STAT3β downregulated genes involved in the cell cycle and PI3K-Akt signaling pathway (Figure 5A,B and Appendix A). These results emphasized that STAT3β inhibited cell proliferation and induced cell death and were consistent with the results of clonogenicity and cell proliferation assays as previous demonstrated. Furthermore, we examined the relationship between differentially expressed genes and the TNF signaling pathway. As Figure 5C shows, 49 TNF signaling genes were upregulated in highly expressed STAT3β ESCC cells, especially under CCRT treatment. This analysis clearly highlighted that STAT3β activated the TNF signaling pathway and TNF was one of inducers of cell necrosis. Thereafter, we verified the effect of STAT3β overexpression on the mRNA expression of TNF signaling pathway genes, including IL1B, MMP9, CXCL2, IL6, IL1A, CASP1, IFI16, MLKL, and RIPK1 in KYSE30-Vector and STAT3β cells after CCRT. As Figure 5D shows, all genes have a high expression in STAT3β cells, especially after CCRT treatment. We then examined the expression of necroptosis-related genes and the cell cycle pathway (Figure 5E). We found that cyclin A2, cyclin B2, and PLK1 were downregulated, and RIPK1 and MLKL were upregulated in STAT3β expression cells after treatment with CCRT. RIPK1 and MLKL are the major mediators of necroptosis [32]. We evaluated the effects of STAT3β on necroptosis under cisplatin and IR treatment using flow cytometry. STAT3β significantly increased cell death under cisplatin or IR alone and combined (Figure 5F). To confirm this result, RIPK1 and MLKL protein levels were increased and the phosphorylation of MLKL was clearly enhanced in STAT3β high-expression KYSE30 and KYSE150 cells after CCRT (Figure 5G–I). The phosphorylation of STAT3α was decreased and there were fewer STAT3α in the nucleus after CCRT in KYSE30 and KYSE150 Vector expressing cells, and STAT3β induced an increase in the expression of pSTAT3α and nuclear retention (Appendix A). To confirm that cellular necroptosis was induced by STAT3β, we used cellular necroptosis inhibitor Necrostatin-1 and (E)-Necrosulfonamide, which blocks necroptosis through the inhibition of RIPK1 and MLKL. As shown in Figure 5J,K, cell viability was increased and cell death was reduced after concurrent chemoradiotherapy with Necrostatin-1 (Nec-1) and (E)-Necrosulfonamide (NSA) in KYSE30. These results demonstrate that STAT3β enhances ESCC cell necroptosis in response to CCRT by upregulating TNF and necroptosis signaling pathway-related proteins.

## 3. Discussion

Esophageal cancer is an aggressive upper gastrointestinal malignancy which presents as a locally advanced tumor that requires multimodal therapy. The prognosis in patients with esophageal cancer remains poor, with a five-year survival of 15–34% [42,43,44]. The treatment options for esophageal cancer include endoscopic resection, curative surgery, and chemoradiotherapy. In recent years, preoperative and postoperative chemotherapy, as well as concurrent chemoradiation, have initially improved the survival time of locally advanced esophageal cancer patients [38,45,46,47]. The RTOG-8501 trial showed that the overall five-year survival rate was 26% with chemoradiotherapy, compared with 0% with radiotherapy alone [48]. However, the clinical outcomes of patients treated with CCRT are different. M. Li et al. [49] reported 56 patients with clinical T4M0 ESCC who received CCRT using involved-filed irradiation (IFI). Only 13 (23.2%) patients achieved a complete response. In our data, 33 (31.4%) patients (*n* = 105) had a complete response to CCRT (Figure 2A), with others partially responding or even progressing during treatment. There is a clinical benefit for patients who experience a significant tumor remission after CCRT, while there is a problem for patients who are not sensitive to CCRT. VEGF kinetics is a prognostic factor for locally advanced ESCC patients receiving CCRT [50]. Y.H. Chen et al. reported that lower post-treatment VEGF levels and decreasing levels of VEGF during CCRT are significantly associated with better clinical outcomes. Therefore, it is crucial to apply molecular markers to predict the treatment response and prognosis in esophageal cancer. 

STAT3 is found to be constitutively active in ESCC and involved in resistance to chemo(radio)therapy [19,51]. The STAT3 isoform STAT3β has been described as a tumor suppressor, which can be a protective prognostic marker in acute myeloid leukemia, especially in ESCC patients with adjuvant chemoradiotherapy [27,29]. Our data further assess cytoplasmic STAT3β as a protective prognostic marker, demonstrating that it is valuable in terms of indicating the ratio of complete response under CCRT (*p =* 0.043) (Figure 2A). It has been demonstrated that a high ratio of STAT3β/STAT3α correlates with favorable clinical prognoses in acute myeloid leukemia [29]. Low-expression nuclear STAT3α is better for clinical prognoses alone or in combination with high-expression cytoplasmic STAT3β (Figure 1F), suggesting the importance of the maintenance of balanced STAT3β/STAT3α expression. The reason for the combined prognostic model may be the inhibition of the transcriptional ability of STAT3α via STAT3β [27]. However, the role of cytoplasmic STATβ has never been reported, thus these results need to be confirmed in larger confirmatory studies. 

In addition, the enforced expression of STAT3β enhances chemosensitivity in ESCC cells in a STAT3β-dose-dependent manner [27]. Furthermore, we describe the role of STAT3β in the exposure of CCRT. High STAT3β expression significantly reduces colony formation with CCRT exposure (Figure 4B,D). Furthermore, the cell proliferation revealed that the number of EdU-positive staining cells mediated by highly expressed STAT3β is small when compared with the number of positive cells mediated by the vector under the same conditions (Figure 4E,F). From these data, we can see that KYSE150- STAT3β is more resistant than KYSE30- STAT3β, because KYSE150- STAT3β needs a higher dose of cisplatin. Although all seven types of ESCC cells that we detected are low STAT3β expression, they demonstrate varying sensitivities to cisplatin (Figure 3B). This may be due to the different backgrounds and other mechanisms involved in cisplatin cytotoxicity resistance, such as their DNA repair ability [52]. Taken together, our data suggest that STAT3β is a tumor suppressor and enhances sensitivity to CCRT.

The transcription activation domain of STAT3 binds to transcription co-activators such as p300/CBP and c-jun [53,54]. The opposing function between STAT3α and STAT3β may be the result of its specific set of target genes. Excepting the dominant-negative regulator of STAT3α, STAT3β can repress the expression of Bcl-xL, p21, cyclin D1, and cyclin C, inducing apoptosis and cell cycle arrest in cancer cells [21,26,55]. STAT3β has been shown to prolong the phosphorylation of STAT3α Y705 and nuclear retention. The heterodimer STAT3α, STAT3β, becomes more stable. STAT3β can cross-regulate or enhance the transcriptional activity of STAT3α [23,56]. In the context of chemoradiotherapy-resistant cells, STAT3 can be activated with cisplatin and ionizing radiation treatment. We found the phosphorylation of Y705 of STAT3 was decreased after cisplatin and ionizing radiation treatment in our KYSE150 and KYSE510 cell lines (Figure 3C). This result suggests that activated STAT3 has an effect on chemoradiotherapy resistance and STAT3β contributes to sensitivity to chemoradiotherapy in ESCC cells.

The role of necroptosis is considered to be two-sided in cancer. On the one hand, necroptosis serves as a supplementary way to induce cell death when tumor cells evade apoptosis to survive. On the other hand, cell necrosis can lead to the release of cell contents, which trigger inflammatory responses and promote cancer metastasis and immunosuppression [57,58]. The expression of several key regulators in the necrotic pathway are downregulated in different types of cancer cells, such as breast cancer, colorectal cancer, and head and neck squamous cell carcinoma [32]. Yulin Sun et al. [59] reported that RIPK3 was downregulated in esophageal cancer and its expression was associated with a better response to chemotherapy and prolonged survival. This demonstrates that necroptosis may be positive in esophageal cancer through regulating cisplatin sensitivity and inducing cell necrosis. There is accumulating evidence of the induction of cellular necroptosis involving STAT3 function of transcription [60,61,62]. In our study, we found high STAT3β expression contributed to cellular necroptosis and cell cycle arrest after CCRT (Figure 5A,B). In highly expressed STAT3β ESCC cells, necroptosis is dramatically activated compared with the empty vector groups after CCRT treatment (Figure 5F). Thus, the sensitivity to CCRT of STAT3β in ESCC cells is dependent on increased cell necroptosis and cell cycle arrest. Alyssa D. Smith et al. [59] reported that the STAT3-DNMT axis silences the TNFα-RIP1 necroptosis pathway in myeloid-derived suppressor cells. DNMT1 and NDMT3b are transcriptionally regulated by STAT3, and activated STAT3 induced DNMT1 expression and hypermethylated TNFα in MDSCs. In the ESCC chemoradiotherapy model, STAT3β upregulated the TNF signaling pathway and necroptosis, which may be a negative factor of STAT3α. 

## 4. Materials and Methods

### 4.1. Patient Samples

This study was approved by the ethical committee of the Central Hospital of Shantou City (2016-026, 4 November 2016) and the ethical committee of Shantou University Medical College (SUMC-2017-12, 1 January 2018). Written informed consent was obtained from all patients. Pretreatment tumor biopsies of CCRT patients were performed as described previously [63]. Moreover, 105 samples were selected, which had received chemoradiotherapy from 2014 to 2016, based on pathologic diagnosis in this study. The clinical information of patients is shown in Table 3. The histologic characteristic and clinicopathologic staging of samples were classified according to the Eighth Edition of American Joint Committee on Cancer Tumor-Nodes-Metastasis staging system. 

### 4.2. Immunohistochemistry

Immunohistochemistry (IHC) was processed according to the protocol described previously [64]. Monoclonal antibodies against STAT3α (Cell Signaling Technology, 9145s, 1:50) and STAT3β (G10H9, 1:500, a gift from Dr. David J. Tweardy) were used [65]. STAT3α and STAT3β in tissue specimens of tumor cells were analyzed and scored according to the intensity of staining and the proportion of positive stained tumor cells. A protein expression score was determined based upon the intensity of staining according to a Vectra automated multispectral histopathological quantitative analysis system (InForm Version 2.1, PerkinElmer). The relationship between protein and outcome was analyzed by X-tile [66]. The patients were classified into two groups: low expression was scored 0 and high expression was scored 1. Finally, the score was calculated as the average of the samples.

### 4.3. Cell Cultures

Human ESCC cell lines KYSE30, KYSE150, KYSE450, and KYSE510 were grown in RPIM 1640 (HyClone, Logan, UT, USA) supplemented with 10% fetal bovine serum (FBS) and penicillin/streptomycin. HEK293T cells were cultured in Dulbecco’s modified Eagle’s medium (HyClone, USA), 10%FBS and 100 mg/mL penicillin/streptomycin. 

### 4.4. Lentivirus Infection

The recombinant vector containing STAT3β cDNAs was constructed by exonuclease III and co-transfected HEK293T cells with packaging plasmids pMDLg/pRRE, pVSV-G, and pRSV-Rev using Lipofectamine 3000 transfection reagent (Life Technology, Carlsbad, CA, USA). Lentivirus was harvested and filtered after 72-h transfection. Cells were infected with lentiviruses in 10 μg/mL polybrene (Santa Cruz Biotechnology, Dallas, TX, USA). The primer sequences used for STAT3β amplifying are as follow: human STAT3β (Forward: 5′-ATGGCCCAATGGAATCAG-3′, Reverse: 5′-TTATTTCCAAACTGCATCAATGAAT-3′).

### 4.5. Western Blotting

Extracts were prepared from cells with Laemmli sample buffer (Bio-Rad, Hercules, CA, USA). Proteins were resolved by sodium dodecyl sulfate polyacrylamide gel electrophoresis and transferred onto polyvinylidene fluoride membrane. The membrane was blocked in 5% non-fat dry milk within Tris-buffered saline Tween-20 and incubated with pSTAT3 (Cell Signaling Technology, 9145s, 1:1000), STAT3 (Cell Signaling Technology, 9139s, 1:1000), RIPK1 (D94C12) XP^®^ Rabbit mAb (Cell Signaling Technology, 2493T, 1:1000), Anti-MLKL (phospho S345) (Abcam, ab196436, 1:1000), MLKL Polyclonal Antibody (Proteintech, 21066-1-AP, 1:1000), Anti-Flag M2 monoclonal antibody (Sigma-Aldrich, St. Louis, MO, USA, F3165, 1:5000), GAPDH (Santa Cruz Biotechnology, sc-47724, 1:1000) and β-Actin (Santa Cruz Biotechnology, sc-47778, 1:1000). Immunoblotting was further performed with horseradish peroxidase-linked anti-mouse (Santa Cruz Biotechnology, sc-516102, 1:5000) or anti-rabbit IgG (Cell Signaling Technology, 7074s, 1:2000) and then photographed using ChemiDoc MP (Bio-Rad, Hercules, CA, USA).

### 4.6. RNA Extraction and Quantitative RT-PCR

The total RNA was purified from cells using TRIzol (Life Technologies, Shanghai, China) followed by reverse transcription to cDNA using HiScript® III RT SuperMix for qPCR (+gDNA wiper) (Vazyme, Nanjing, China, R323-01) according to the manufacturer’s instruction. Quantitative RT-PCR was performed using ChamQ Universal SYBR qPCR Master Mix (Vazyme, Nanjing, China, Q711-02) and Applied Biosystems 7500/7500 Fast Real-Time PCR System (Thermo Fisher, Shanghai, China). The primers sequences used for quantitative RT-PCR are provided in Appendix A.

### 4.7. Clonogenic Assay

For the experiment groups, cells were treated with different concentrations of cisplatin (0 μM, 0.25 μM, 0.5 μM, and 1 μM) for 24 h, and different doses of radiation (0 Gy, 1 Gy, 2 Gy, and 4 Gy) using an X-ray irradiator (RS2000PRO). The treatments were performed alone and in combination. Cells were harvested after treatment with chemo(radio)therapy. Cells were counted. The number ranged from 200 to 500. They were seeded into 12-well plates and further incubated for 8–10 days. The colonies were fixed with a mixture of methanol and acetic acid for 15 min and stained with 0.5% crystal violet for 15 min. Colony numbers were counted using a microscope and an automatic counting tool in ImageJ. 

### 4.8. Cell Viability Assay

Cell viability was determined using CellTiter 96^®^ Aqueous one solution cell proliferation assay (Promega Corporation, Beijing, China) according to the manufacturer’s instructions. A total of 10,000 cells were seeded into 96-well plates. Cells were treated with different concentrations of cisplatin (0, 5, 10, 25, 50, and 100 μM) for 24 h. After incubation with tetrazolium compound [3-(4,5-dimethylthiazol-2-yl)-5-(3-carboxymethoxyphenyl)-2-(4-sulfophenyl)-2H-tetrazolium, inner salt; MTS] at 37 °C for 2 h, the results were taken by enzyme-labeled instrument (Multiskan MK3, Thermo Fisher) at 492 nm.

### 4.9. Cell Proliferation Assay 

Cells were seeded on the cell coverslip and treated with or without cisplatin and radiation. After 24 h incubation, cells were labeled with EdU for 4 h. The media was removed and fixed using 4% paraformaldehyde in PBS for 15 min followed by 0.3% Triton X-100 permeabilization reagents for 10 min at room temperature. Click additive solution was prepared using a BeyoClickTM EdU cell proliferation kit with Alexa Fluor 488 (Beyotime Biothechnology, Shanghai, China) according to the instructions. This was added to each well containing a coverslip and incubated for 30 min at room temperature, protected from light. For nuclear staining, diluted Hoechst 33342 was added to the coverslip for 5 min at room temperature and protected from light. This was then imaged and analyzed using a Zeiss 800 confocal microscope (Zeiss, Jena, Germany). The numbers of proliferative cells (EdU-positive, Alexa fluor 488) were counted in six random fields of view per slide. The number of cells in each field of view is in the range of 50-500. EdU-positive cells and Hoechst 33342-positive cells were calculated using Image J. EdU positive cells (%) = EdU-positive cells / Hoechst 33342-positive cells *100.

### 4.10. Cell Death Assay

KYSE30 and KYSE150 cells stably expressing empty vector or STAT3β were seeded into 12-well plates and treated as indicated. Cell deaths were assessed using an Annexin V-FITC apoptosis detection kit (#C1062M, Beyotime, Shanghai, China). In brief, cells were trypsinized and resuspended with Binding Buffer containing PI and Annexin V and quantified on a BD Accuri™ C6 flow cytometer. 10,000 cells were counted per sample. PI-positive cells were counted as death cells.

### 4.11. RNA-seq and Bioinformatics

KYSE30 cells stably expressing empty vector or STAT3β were seeded into 6-well plates and treated with cisplatin (0.2 μM). After 24 h, 5Gy radiation was applied alone or combined with cisplatin treatment. Total RNA was extracted using the TRIzol (Life Technologies, Shanghai, China) method. RNA was subjected to library construction and RNA-seq analyses were performed on the BGISEQ-500 system by BGI (Wuhan, China). Data were aligned to genomes using STAR (version 2.7.6a) and differentially expressed mRNAs were identified by DESeq2 (version 1.16.1). A fold change cutoff of log2 < −0.5 or > 0.5 and a *p*-value cutoff of *p <* 0.05 were deemed significant for the regulated gene sets. Enrichment analysis of Gene Ontology (GO) functional and Kyoto Encyclopedia of Genes and Genomes (KEGG) analysis were performed to identify potential biological processes and pathways through the Metascape [67].

### 4.12. Nuclear Plasma Separation 

KYSE30 and KYSE150 cells stably expressing empty vector or STAT3β were seeded into 10 cm dish and treated with cisplatin and radiation. Cells were washed three times with PBS and lysed using nuclei extraction buffer (NEB) (0.01 M Tris-HCl pH 8.0, 0.01 M NaCl, 0.003 M MgCl2, 0.03 M Sucrose, 0.5% NP-40) with a protease inhibitor cocktail. Cell lysate was centrifuged at 16,000× *g* for 5 min at 4 °C, the supernatant as cytoplasm fraction and the precipitate as nuclear fraction.

### 4.13. Immunofluorescence (IF)

KYSE150 STAT3-KO cells were seeded onto coverslips for 24 h. Cells were transfected with Flag-STAT3α, STAT3β-HA, or a combination of both. After transfection, cells were fixed with 4% paraformaldehyde for 10–15 min, and then washed three times with PBS. Then, 0.1% Triton X-100 permeabilization reagents was added for 10 min. Cells were washed and blocked in 5% normal donkey serum for 1 h. Then, cells were incubated with anti-Flag M2 mouse monoclonal antibody (F3165, Sigma) and anti-HA (sc-7392, Santa cruz) overnight, and cells were stained with Alexa Fluor 488-conjugated donkey anti-mouse IgG (H + L) (1:500, 715-545-150, Jackson) and Alexa Fluor^®^ 647 AffiniPure donkey anti-rabbit IgG (H + L) (1:500, 711-605-152, Jackson) secondary antibodies for 60 min at RT, then washed three times with PBS and counterstained with DAPI (#D9564-10MG, Sigma) at RT for 10 min to visualize the nuclei. Images were obtained and processed by laser-scanning confocal microscopy (LSM800, Carl Zeiss).

### 4.14. Statistical Analysis

Statistical analyses were performed using SPSS V.19 (IBM, Chicago, IL, USA) and GraphPad Prism 7 (La Jolla, CA, USA). All experiment data were described as the mean ± SEM. Independent groups of samples were evaluated using Student’s *t*-test. Survival curves were analyzed using the Kaplan–Meier method with log-rank test. Analysis of the associations between patient survival and clinicopathologic parameters was performed with the Cox proportional hazards models. The *p* value was two-sided, and a *p* value of <0.05 was considered statistically significant.

## 5. Conclusions

The present study demonstrates STAT3β to be a favorable prognostic marker. It acts as a tumor suppressor in ESCC and significantly enhances sensitivity to chemoradiotherapy, and thus can be used as a biomarker for CCRT response prediction. Our study provides novel evidence that STAT3β induces cellular necroptosis upon CCRT exposure and specifically regulates gene expression, including the TNF signaling pathway and necroptosis-associated genes.

## Figures and Tables

**Figure 1 cancers-13-00901-f001:**
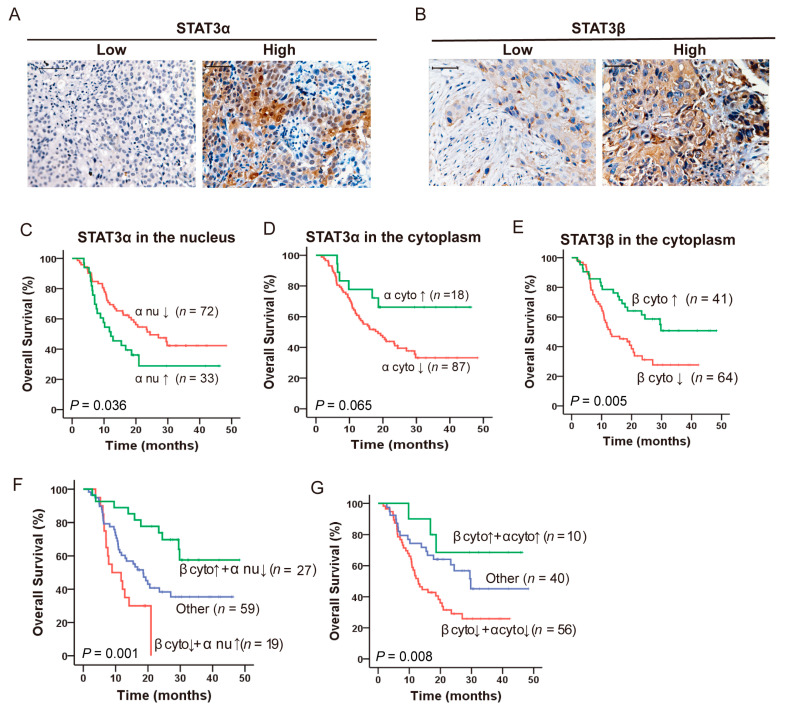
Expression and prognosis significance of STAT3α and STAT3β in ESCC. (**A**,**B**) Representative low and high expression of STAT3α (**A**) and STAT3β (**B**) were analyzed by immunohistochemistry (IHC) in 105 ESCC samples with concurrent chemoradiotherapy (CCRT). Scale bar, 50 μm. (**C**) Kaplan–Meier survival curve of nucleus (nu) STAT3α expression on overall survival in ESCC patients with CCRT. (**D**) Survival curve of cytoplasmic (cyto) STAT3α in ESCC patients with CCRT. (**E**) Survival curve of cytoplasmic (cyto) STAT3β in ESCC patients with CCRT. (**F**,**G**) Survival curves of different expression of nucleus or cytoplasmic STAT3α associated with cytoplasmic STAT3β in ESCC patients with CCRT. ‘↑’ means high expression, ‘↓’ means low expression.

**Figure 2 cancers-13-00901-f002:**
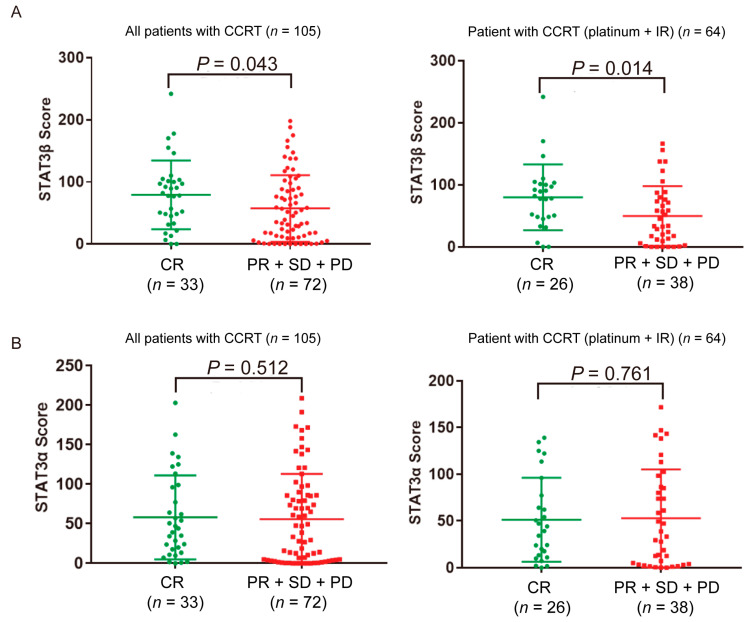
The correlation between STAT3β and STAT3α score and the efficacy of CCRT. Paraffin-embed biopsy sample from 105 ESCC patients who received concurrent chemoradiotherapy (64 patients who received platinum plus ionizing radiation) were analyzed. The expression of each protein in tissue specimens was evaluated and digitized based upon the intensity of staining. (**A**) STAT3β expression in CR compared to PR + SD + PD in all patients with CCRT (**left**) and platinum + IR (**right**). (**B**) STAT3α expression in CR compared to PR + SD + PD in all patients with CCRT (**left**) and platinum + IR (**right**). CR, complete response; PR, partial response; SD, stable response; PD, progressive disease; IR, ionizing radiation.

**Figure 3 cancers-13-00901-f003:**
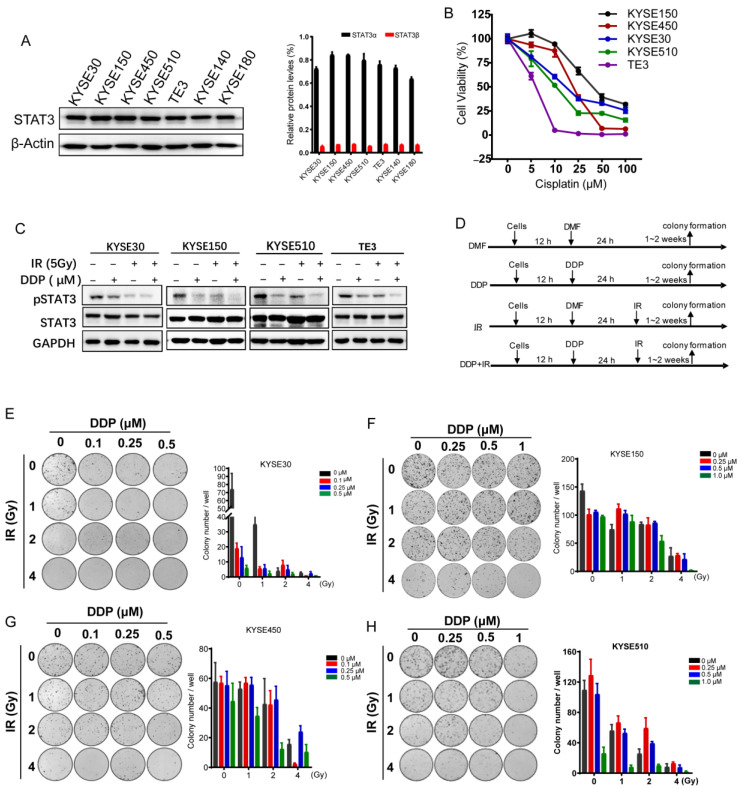
Sensitivity to cisplatin/ionizing radiation chemoradiotherapy in ESCC cells. (**A**) The characteristics of STAT3 in esophageal cancer cell lines. Left, the expression of STAT3 is detected by Western blotting. Right, the histogram of gray value of STAT3α and STAT3β. Black represents STAT3α; red represents STAT3β. The uncropped Western Blot Figures in Appendix A. (**B**) The line chart exhibits cell viability after 24 h of treatment with different concentrations of cisplatin (DDP) in five ESCC cell lines. (**C**) KYSE150 and KYSE510 were treated with DDP (0.5 μM) or ionizing radiation (IR, 5 Gy) or a combination of DDP with IR (DDP + IR). The uncropped Western Blot Figures in Appendix A. (**D**) The flow chart of the colony formation assay under DDP/IR chemoradiotherapy treatment. DMF: N, N-Dimethylformamide. (**E**–**H**) The results of the colony formation of KYSE30, KYSE150, KYSE450, and KYSE510, respectively. Left/upper, the typical diagrams of colony formation. Right, the statistical histogram of numbers of colony formation. Error bars represent means ± SEM.

**Figure 4 cancers-13-00901-f004:**
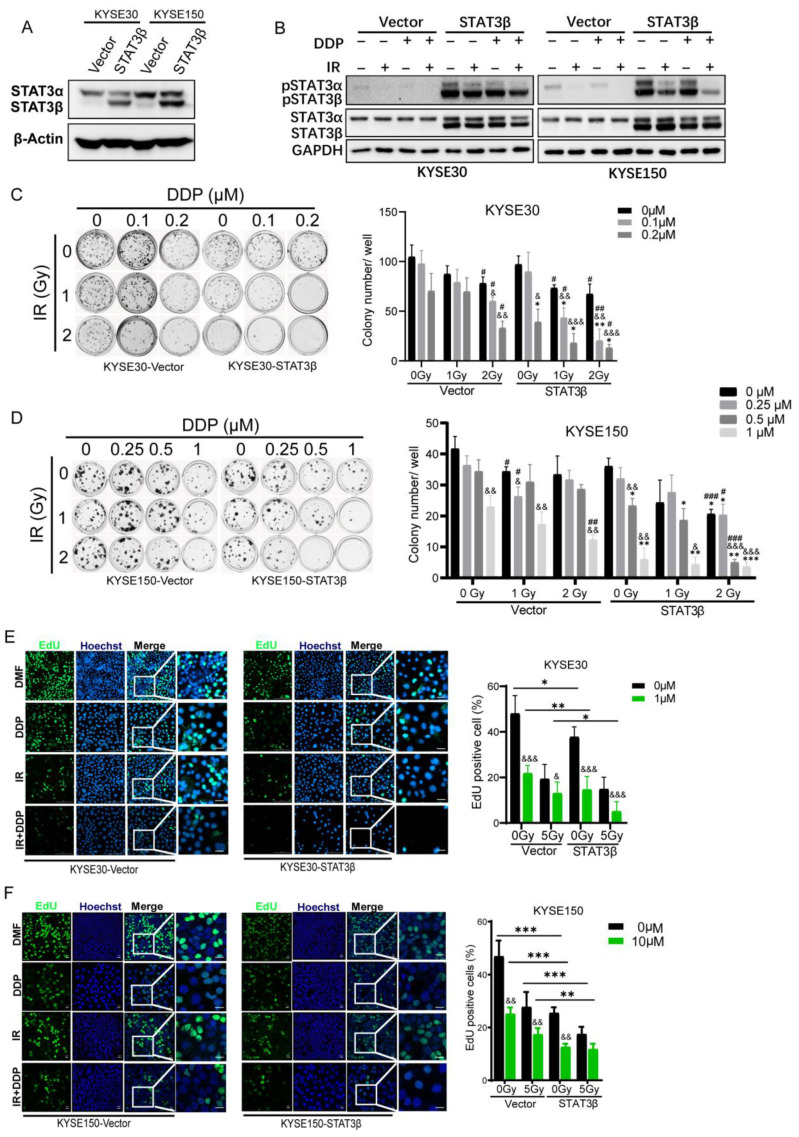
Highly expressed STAT3β enhances the sensitivity to chemoradiotherapy in ESCC cells. (**A**) Immunoblotting showed the high expression of Vector and STAT3β in KYSE30 and KYSE150. The uncropped Western Blot Figures in Appendix A. (**B**) Immunoblotting analysis. KYSE30 (**left**) and KYSE150 (**right**) overexpressing Vector or STAT3β following treatment with vehicle, DDP (0.5 μM), IR (5 Gy), and DDP+IR. The uncropped Western Blot Figures in Appendix A. (**C**,**D**) The ability of colony formation of highly expressed empty vector and STAT3β under DDP/IR chemoradiotherapy treatment. Left, the representative diagrams of colony formation. Right, statistical histogram of colony formation number. Error bars represent mean ± SEM of three independent experiments. (**E**,**F**) The results of EdU assay. KYSE30 (**E**) and KYSE150 (**F**). Left, immunofluorescence diagrams, scale bar, KYSE30 (200 μm), KYSE150 (20 μm). Right, the statistical histogram of EdU positive cells. DDP represents cisplatin, IR represents ionizing radiation. Value is mean ± SEM, ^#^
*p <* 0.05, ^##^
*p <* 0.01, ^###^
*p <* 0.001 compared with 0 Gy at the same concentration of DDP; ^&^
*p <* 0.05, ^&&^
*p <* 0.01, ^&&&^
*p <* 0.01 compared with 0 μM DDP at the same dose of IR; * *p <* 0.05, ** *p <* 0.01, *** *p <* 0.001 compared with the same Vector treatment. Student’s *t*-test was used for all statistical testing.

**Figure 5 cancers-13-00901-f005:**
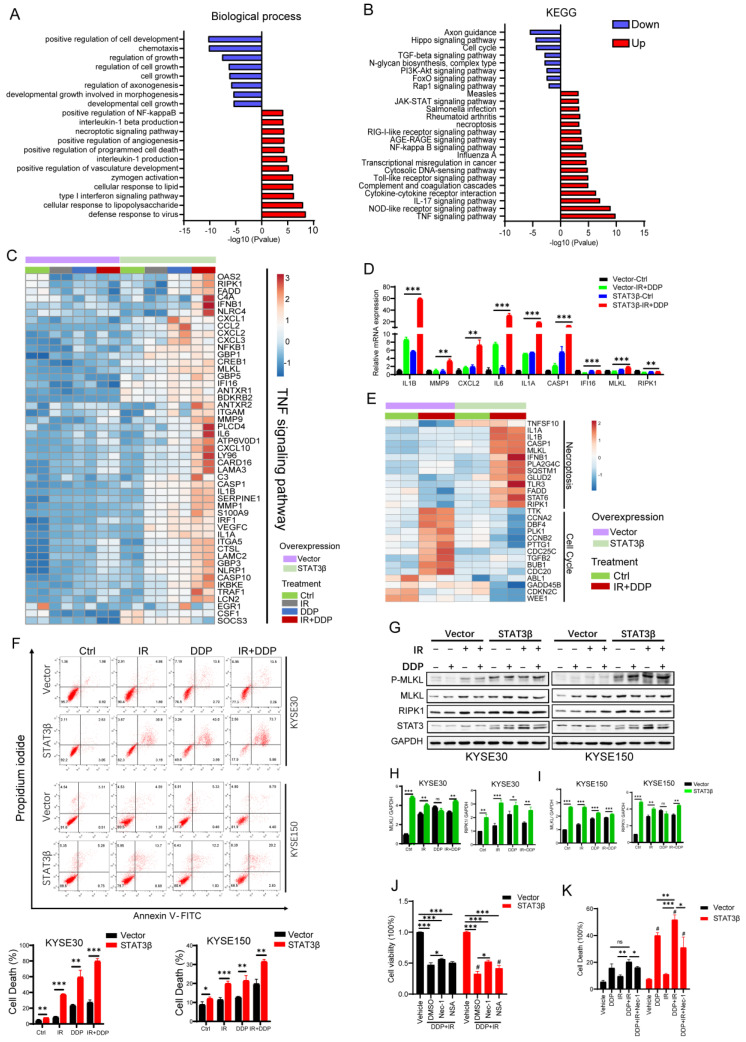
STAT3β overexpression-enhanced cell necroptosis after chemoradiotherapy. (**A**) The gene ontology (GO) enrichment analysis of differentially expressed genes (DEGs) (STAT3β vs. Vector upon chemoradiotherapy). (**B**) The KEGG pathways enrichment analysis of DEGs. The left side of abscissa 0, the enrichment of downregulated genes. The right side of abscissa 0, the enrichment of upregulated genes. (**C**) Heatmap of 49 DEGs in the TNF signaling pathway. KYSE30 overexpressing Vector and STAT3β were treated with Vehicle, DDP, IR, and DDP+IR, in eight groups, two in parallel. (**D**) The mRNA expression of TNF signaling pathway genes by qualitative RT-PCR. Error bars represent means ± SEM, * *p <* 0.05, ** *p <* 0.01, *** *p <* 0.001. (**E**) Heatmap of cell cycle and necroptosis pathway. (**F**) Propidium iodide (PI) and Annexin V double staining was used to determine cell death by flow cytometry. The results of three independent experiments are presented on the left. (**G**) Immunoblotting shows the RIPK1, MLKL, and the phosphorylation of MLKL expression in KYSE30 (left) and KYSE150 (right), following the indicated treatment. The uncropped Western Blot Figures in Appendix A. (**H**,**I**) The histogram of gray value of MLKL and RIPK1 in KYSE30 (**H**) and KYSE150 (**I**). (**J**) The Bar graph exhibits cell viability after 24 h treatment with Vehicle, DDP+IR, DDP + IR + Nec-1 (10μM), DDP + IR + NSA (1μM). (**K**) KYSE30 cells were treated with Vehicle, DDP, IR, DDP + IR and DDP + IR + Nec-1 (10μM) for 24 h; cells were harvested and cell death was measured. * *p <* 0.05, ** *p <* 0.01, *** *p <* 0.001. ^#^
*p <* 0.05 compared to the same vector treatment.

**Table 1 cancers-13-00901-t001:** Univariate and multivariate analysis of STAT3 subtypes associated with overall survival (OS) of esophageal squamous cell carcinoma (ESCC) patients with concurrent chemoradiotherapy (*n* = 105)

Variables	Univariate Analysis		Multivariate Analysis	
	OS		OS	
	HR (95% CI)	*p*	HR (95% CI)	*p*
Age (>64.3 vs. ≤64.3)	1.004 (0.605 to 1.666)	0.988		
Gender (Female vs. Male)	0.995 (0.526 to 1.884)	0.988		
cTNM classification		0.435		
III vs. II	0.806 (0.377 to 1.723)	0.578		
IV vs. III	1.188 (0.598 to 2.358)	0.623		
Chemoradiotherapy regimen				
(Others vs. Platinum + IR)	1.553 (0.933 to 2.586)	0.090	1.572 (0.944 to 2.618)	0.082
STAT3 β in the cytoplasm (High vs. Low) ^a^	0.458 (0.261 to 0.802)	0.006 **	0.424 (0.241 to 0.748)	0.003 **
STAT3 α in the nucleus (High vs. Low) ^b^	1.769 (1.039 to 3.011)	0.036 *	1.937 (1.127 to 3.328)	0.017 *
STAT3 α in the cytoplasm (High vs. Low) ^c^	0.460 (0.198 to 1.071)	0.072		

Note: Cox proportional hazards regression model. Variables were adopted for their prognostic significance by univariate analysis. OS: overall survival; IR: ionizing radiation. ^a^ low, ≤76.9 scores; high, >76.9 scores; ^b^ low, ≤77.3 scores; high, >77.3 scores; ^c^ low, ≤126 scores; high, >126 scores. * *p* < 0.05, ** *p* < 0.01.

**Table 2 cancers-13-00901-t002:** Correlation between clinical parameters and STAT3β subtypes in ESCC patients with concurrent chemoradiotherapy (*n* = 105).

Parameters	Patient Number	Expression of STAT3β in Tumor Cytoplasm		R	*p* *
		High	Low ^a^		
Gender					
Male	85	30 (35.3%)	55 (64.7%)	0.159	0.104
Female	20	11 (55%)	9 (45%)		
Age					
≤64.3 years	51	21 (41.2%)	30 (58.8%)	−0.042	0.664
>64.3 years	54	20 (37.0%)	34 (63%)		
Response					
CR	33	19 (57.6%)	14 (42.4%)	−0.257	0.008
PR + SD + PD	72	22 (30.6%)	50 (69.4%)		
Chemoradiotherapy regimen					
Platinum + IR	64	25 (39%)	39 (61%)	0.000	0.997
Others	41	16 (39%)	25 (61%)		
cTNM classification					
II	19	6 (31.6%)	13 (68.4%)	0.029	0.664
III	34	15 (44.1%)	19 (55.9%)		
IV	52	20 (38.5%)	32 (61.5%)		

* Chi-Square value; *p* < 0.05 was considered significant. All patients received concurrent chemoradiotherapy (CCRT). CR: complete response; PR: partial response; SD: stable response; PD: progressive disease; IR: ionizing radiation. ^a^ Low, ≤76.9 scores; high, >76.9 scores.

**Table 3 cancers-13-00901-t003:** Characteristics of the ESCC patients with CCRT

Parameters	Total Patient Number (*n* = 105)		1-Year OS	2-Year OS	*p* *
	Patient Number	Percentage			
Gender					
Male	85	81%	64.7%	40.5%	0.998
Female	20	19%	60.0%	54.5%	
Age					
≤64.3 years	51	48.6%	66.7%	44.6%	0.988
>64.3 years	54	51.4%	63.0%	44.6%	
cTNM Classification					
II	19	18.1%	68.4%	41.4%	0.432
III	34	32.4%	67.6%	55.1%	
IV	52	49.5%	61.5%	38.2%	
Response					
CR	33	31.4%	97%	83.5%	0.000
PR	64	61%	51.6%	27.4%	
SD	6	5.7%	33.3%	16.7%	
PD	2	1.9%	50%		
Chemoradiotherapy Regimen					
Platinum + IR	64	61%	68.8%	50.1%	0.088
Others	41	39%	58.5%	35.6%	

* Chi-Square value; *p <* 0.05 was considered significant. All patients received concurrent chemoradiotherapy (CCRT); CR: complete response; PR: partial response; SD: stable response; PD: progressive disease; OS: overall survival; IR: ionizing radiation.

## Data Availability

All data presented in this study are included within the paper and its Supplementary files.

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
