# Peer review of "STAT3β Enhances Sensitivity to Concurrent Chemoradiotherapy by Inducing Cellular Necroptosis in Esophageal Squamous Cell Carcinoma"

_cancers, 2021, doi:10.3390/cancers13040901_

Round 1

Reviewer 1 Report

Thanks to the authors for the detailed revised manuscript of STAT3β enhances sensitivity to concurrent chemoradiotherapy by inducing cellular necroptosis in esophageal squamous cell carcinoma. Most of the comments can be addressed  in revised manuscript. Some minor revisions can be modified.

1) Some references should be cited in line 327 to support the possible mechanisms of cisplatin resistance (for example, Barry et. al., 2010 and/or Sugase et. al., 2017).

2) The room-in pictures of Fig. 4E and F also need a scale bar.

3) The size of the counted fields should add in section 4.9 and 4.10.

4) The unit of scale bar in Fig. S1 is μm.

Author Response

1) Some references should be cited in line 327 to support the possible mechanisms of cisplatin resistance (for example, Barry et. al., 2010 and/or Sugase et. al., 2017).

We have now included the requested reference in line 327.

2) The room-in pictures of Fig. 4E and F also need a scale bar.

We have added scale bar in room-in pictures.

3) The size of the counted fields should add in section 4.9 and 4.10.

In section 4.9, all cells in a random field of view are counted and the number of cells in each field of view is in the range of 50-500. In section 4.10, 10,000 cells were counted per sample by C6 flow cytometer. The size of the counted fields is added in section 4.9 and 4.10.

4) The unit of scale bar in Fig. S1 is μm.

This error has now been corrected.

Reviewer 2 Report

With these corrections the manuscript is suitable for publication

Author Response

Thank you for the time you have spent reviewing our manuscript.

Reviewer 3 Report

I appreciate the response to my points.

Line 199 should be „intracellular“ instead of “intercellular”.

The new paragraphs requrire some small grammar editing.

Author Response

1) Line 199 should be “intracellular” instead of “intercellular”.

Thank you for your careful work. We have corrected the mistake.

2) The new paragraphs require some small grammar editing.

We have made this requested change. In line 471, "Then, 0.1% Triton X-100 was added for 10 min for permeation. " changed to "Then, 0.1% Triton X-100 permeabilization reagents was added for 10 min.".

This manuscript is a resubmission of an earlier submission. The following is a list of the peer review reports and author responses from that submission.

Round 1

Reviewer 1 Report

This manuscript entitled “STAT3β enhances sensitivity to concurrent chemoradiotherapy by inducing cellular necroptosis in esophageal squamous cell carcinoma” by Zheng et al. reported that STAT3β expression improves the efficacy of concurrent chemoradiotherapy through necroptosis by TNF signaling pathway. The relationships between STAT3β expression and sensitivity to treatment in ESCC patients and cell lines were entirely analyzed. But the concept of cellular necroptosis induced by STAT3β overexpression was not detailed investigated in ESCC cell lines. If the authors want to clarify the effective concurrent chemoradiotherapy that mediates by necroptosis, then they should not just focus on the capacity of cell proliferation and apoptosis by EdU labeling and Annexin V assay. More description and clarification of necroptosis have to be done in this manuscript.

Other comments
1.      The high expression of STAT3β in ESCC patients showed longer overall survival with CCRT, but all of ESCC cell lines exhibited low expression of this isoform that had different sensitivity of cisplatin. The authors should mention this in the section of the discussion.
2.      The range of cisplatin (DDP) dosages in this manuscript from 0.1 to 100 μM (in Fig. 3B), please explain the reasons for different conditions.
3.      This manuscript focused on STAT3β expression, but some of the data not enough clear to distinguish from STAT3 or pSTAT3 staining by Western Blot. Why did the authors not to exam the STAT3α and STAT3β expression by fluorescence staining that same as done in their previous study (Zhang et. al., 2016)?

  1.     Line 190. Fig. 3I is shown in the text but not in the figure.
    5.      The authors mention the resistance of chemoradiotherapy in the introduction, but there had limited data to illustrate in the section of result and discussion. The comparison of the STAT3β overexpression between KYSE30 and 150 should be mention.
  2.     The bar graph in Fig. 4D had some mistakes, please correct it.
    7.      Fig. 4E, F: Could the authors provide higher magnification views pictures? And describe why the different sizes of nuclear staining by Hoechest? Finally, the method in these two figures that how to calculate and statistics the positive EdU labeling cell should be described in the method section.
  3.    Fig. 5 displayed the RNA-seq results from STAT3β overexpression and concurrent chemoradiotherapy, the authors should more explain why focused on the TNF and necroptosis in the text.
  4.    Fig. 5F represented the increased cell death after chemoradiotherapy even in the control group. But there had significant increased (p<0.001) MLKL and RIPK1 expression between vector and STAT3β overexpression in the control group (Fig. 5 G-I), please explanation of the possible influences of STAT3β overexpression in the aspect of cell death.

  1. Line 26: associated with prognosis?
  2. Line 28: ESCC patients with high STAT3β expression tend to have a complete response to concurrent chemoradiotherapy, this sentence seems strange
  3. Line 29 exhibit CCRT sensitivity?

  1. Line 34 who are not suitable for surgery. Not correct
  2. Line 37 response for adjuvant chemoradiotherapy (ACRT) marker ?
  3. Line 42 protection factor ?
  4. STAT3α was described in summary but not in abstract
  5. Line 78 pathogen infections?
  6. Line 93 sensitizes to CCRT ?
  7. The order of tables is wrong.
  8. Almost half patients have stage IV ?
  9. platinum plus ionizing radiation is not the standard regimen for esophageal patients

  1. Figure 2. Do the authors show cytoplasmic STAT3β / nuclear STAT3α or total STAT3β/α in WB as the authors claim the isoform and location of STAT3 may play different role ? In vitro study did no separately cytoplasmic and nuclear STAT3.

  1. Figure 3A: STAT3α and STAT3β expressions were noted found by immunoblotting.

  1. Figure 3B after 24 hours treatment?

  1. Figure 3C decreased pSTAT3 ?

  1. Figure 1A: did not show nuclear and cytoplasmic STATα / STAT3β

Author Response

Response to Reviewer 1 Comments

Major Ponits

Point 1: This manuscript entitled “STAT3β enhances sensitivity to concurrent chemoradiotherapy by inducing cellular necroptosis in esophageal squamous cell carcinoma” by Zheng et al. reported that STAT3β expression improves the efficacy of concurrent chemoradiotherapy through necroptosis by TNF signaling pathway. The relationships between STAT3β expression and sensitivity to treatment in ESCC patients and cell lines were entirely analyzed. But the concept of cellular necroptosis induced by STAT3β overexpression was not detailed investigated in ESCC cell lines. If the authors want to clarify the effective concurrent chemoradiotherapy that mediates by necroptosis, then they should not just focus on the capacity of cell proliferation and apoptosis by EdU labeling and Annexin V assay. More description and clarification of necroptosis have to be done in this manuscript.

Response 1: We thank the reviewer for his / her positive and constructive assessment of our manuscript. We agree with the reviewer that this is an important point that needed to be addressed. To confirm cellular necroptosis induced by STAT3β, we use cellular necroptosis inhibitor Necrostatin-1 and (E)-Necrosulfonamide which blocks necroptosis by the inhibition of RIPK1 and MLKL. The experiment data were shown in Figure 5J, K, cell viability was increased and cell death was reduced after concurrent chemoradiotherapy with Necrostatin-1 (Nec-1) and (E)-Necrosulfonamide (NSA) in KYSE30 cells. It was convincingly demonstrated that cellular necroptosis induced by STAT3β after concurrent chemoradiotherapy. These statements have been added to lines 263 to 267 of the revised manuscript.

Fig. 5 STAT3β overexpression enhanced cell necroptosis upon chemoradiotherapy. (J) The Bar graph exhibits cell viability after 24-h treatment with Vehicle, DDP+IR, DDP + IR + Nec-1 (10μM), DDP + IR + NSA (1μM). (K) KYSE30 cells were treated with Vehicle, DDP, IR, DDP + IR and DDP + IR + Nec-1 (10μM) for 24 h; cells were harvested and cell death was measured. *P < 0.05, **P < 0.01, ***P < 0.001. #P < 0.05, ##P < 0.01, ###P < 0.001 compared to the same vector treatment.
Other comments
Point 1: The high expression of STAT3β in ESCC patients showed longer overall survival with CCRT, but all of ESCC cell lines exhibited low expression of this isoform that had different sensitivity of cisplatin. The authors should mention this in the section of the discussion.
Response 1: We thank the reviewer for this suggestion. Indeed, we found STAT3β exhibited low expression pattern in all ESCC cell lines we have. The mechanism of cisplatin cytotoxicity is the formation of platinum-DNA adducts. Attenuating this type of DNA damage are key factor to decide the sensitivity of cisplatin. Drug import / export, drug activity, DNA damage repair and cell death signaling are related to cisplatin cytotoxicity [1]. Different mechanisms involved in the effects of cisplatin sensitivity because all of ESCC cell lines came from different backgrounds. And we have mentioned it in the section of the discussion (line325-327, “Although all seven types of ESCC cells … such as their DNA repair ability.”). 
Point 2: The range of cisplatin (DDP) dosages in this manuscript from 0.1 to 100 μM (in Fig. 3B), please explain the reasons for different conditions.
Response 2: Thank you for your careful work. The dosages of DDP in Fig. 3B was 0, 5, 10, 25, 50, 100 μM in order to determine the IC50 of cisplatin from 5 types of ESCC cell lines. Different cell types have different sensitivity to cisplatin, and the range of cisplatin dosages were used to screen the DDP IC50 of ESCC cell lines. In addition to different cell types, different density of cell inoculation showed different sensitivity to cisplatin. In clonogenic assay, we found that cisplatin inhibits clone formation at very low doses (0.1-1μM) because individual cells are more sensitive to cisplatin. In EdU assay, the number of EdU positive cells decreased significantly at 1-10μM cisplatin, because low doses of cisplatin inhibited DNA replication. 

Point 3: This manuscript focused on STAT3β expression, but some of the data not enough clear to distinguish from STAT3 or pSTAT3 staining by Western Blot. Why did the authors not to exam the STAT3α and STAT3β expression by fluorescence staining that same as done in their previous study (Zhang et al., 2016)?

Response 3: We would like to thank the reviewer for the suggestions to our manuscript. In Zhang et al. study, STAT3β-overexpression induced high level of pSTAT3α Y705 and co-localization with STAT3α after OSM treatment which proved by confocal microscopy analysis. In our study, for examining the STAT3α and STAT3β expression by fluorescence staining, we constructed stably expressed Flag-STAT3α and STAT3β-HA alone or combined in KYSE150 STAT3-KO cells, and detecting Flag and HA tags to determine their location. STAT3α and STAT3β were co-localization and distributed in nucleus and cytoplasm. These data have been added to lines 198 to 202 of the revised manuscript.

Figure S1. A. KYSE150 STAT3-KO cells transfected with Flag-STAT3α, STAT3β-HA and Flag-STAT3α plus STAT3β-HA, the protein expression were detected by western blotting. B. The location of STAT3α and STAT3β was detected by staining Flag and HA. Scale bar: 10μM.

Point 4: Line 190. Fig. 3I is shown in the text but not in the figure. 

Response 4: We thank the reviewer for pointing this mistake out. In response to the above problems, we have deleted Line 190. Fig. 3I and corrected the Figure legend of Figure 3. 

Point 5: The authors mention the resistance of chemoradiotherapy in the introduction, but there had limited data to illustrate in the section of result and discussion. The comparison of the STAT3β overexpression between KYSE30 and 150 should be mention.

Response 5: We thank the reviewer for this question. In Figure. 4B, D, E, F, we investigated the sensitivity to cisplatin between overexpressing empty vector and STAT3β via clonogenic assay and EdU staining in KYSE30 and KYSE150. From these data, we can see that KYSE150- STAT3β is more resistant than KYSE30- STAT3β, because KYSE150- STAT3β needs a higher dose of cisplatin. And we will mention it in the Discussion (line323-325, “From these data, we can see … needs a higher dose of cisplatin.”)

Point 6: The bar graph in Fig. 4D had some mistakes, please correct it.

Response 6: We thank the reviewer for pointing this mistake out. We have corrected Fig. 4D accordingly.

Point 7: Fig. 4E, F: Could the authors provide higher magnification views pictures? And describe why the different sizes of nuclear staining by Hoechest? Finally, the method in these two figures that how to calculate and statistics the positive EdU labeling cell should be described in the method section. 

Response 7: We apologize for the poor image quality. We have made several changes to improve the quality of our images. We remade all microscopy figures, linearly adjusting brightness and contrast (if possible) and exporting figures in a different file format that improved image quality. And we showed a zoom-in pictures in Fig. 4E, F.
In Figure. 4E, F, KYSE30 and KYSE150 were treated with cisplatin (DDP) or /and ionizing radiation, which results in DNA damage and cell death. The question about the different sizes of nuclear staining is due to cell death after chemoradiotherapy treatment. During programmed death, the cytoplasm is dehydration and condensed lead to cell size decreases. In nuclear, the chromatin shrinks, staining deepens and volume increases, which making the nucleus appear larger. 
The method of calculation and statistics of the positive EdU labeling cell is shown below. The numbers of proliferative cells (EdU-positive, Alexa fluor 488) were counted in six random fields of view per slide. EdU-positive cells and Hoechst 33342-positive cells were calculated using Image J. EdU positive cells (%) = EdU-positive cells / Hoechst 33342-positive cells *100. And we also have described in the Method 4.9 Cell proliferation assay. 

Point 8: Fig. 5 displayed the RNA-seq results from STAT3β overexpression and concurrent chemoradiotherapy, the authors should more explain why focused on the TNF and necroptosis in the text. 

Response 8: We thank the reviewer for this question. According to the results of RNA-seq, TNF signaling pathway is the most significant up-regulated pathway in KEGG. And we found STAT3β overexpression cells were sensitivity to CCRT and associated with cell death in our experiments. Induction of cellular necroptosis is a promising strategy for the treatment of tumors, and necroptosis pathway also enrich in our RNA-seq results, so we look further into these two pathways in our STAT3β overexpression and CCRT model. As suggested, we have added relevant content in the text (line. 240-242, “The TNF signaling pathway is the most …enrich in our RNA-seq results.”)

Point 9: Fig. 5F represented the increased cell death after chemoradiotherapy even in the control group. But there had significant increased (p<0.001) MLKL and RIPK1 expression between vector and STAT3β overexpression in the control group (Fig. 5 G-I), please explanation of the possible influences of STAT3β overexpression in the aspect of cell death. 

Response 9: We apologize to the reviewer if this aspect of the manuscript was not clearly explained. In Fig. 5G, we found MLKL and RIPK1 were increased expression after chemoradiotherapy, and STAT3β overexpression could also upregulate the expression of MLKL and RIPK1. STAT3-DNMT axis silences the TNFα-RIP1 necroptosis pathway in myeloid-derived suppressor cells. [2]. In ESCC cell lines, pSTAT3 was decreased by chemoradiotherapy and STAT3β as a dominant-negative regulator of STAT3α, may represses the expression of STAT3α target genes and affects MLKL and RIPK1 expression from epigenetic level. Necroptosis occurs in response to TNF superfamily receptor or TLR ligands. Taken together, STAT3β may contribute to necroptosis by inducing the expression of key components of necroptosis pathway and enhancing upstream activation signal. These statements have been added to lines 355 to 359 of the revised manuscript.

Point 10: Line 26: associated with prognosis?

Response 10: I’m sorry we didn’t make it clear. In our results, we found that combining cytoplasmic STAT3β and nuclear STAT3α for assessing the correlation with overall survival, high-expression cytoplasmic STAT3β and low-expression nuclear STAT3α is the best combination to prolong overall survival (P=0.001, Fig. 1F). We have deleted it in our text.

Point 11: Line 28: ESCC patients with high STAT3β expression tend to have a complete response to concurrent chemoradiotherapy, this sentence seems strange

Response 11: We thank the reviewer for pointing this out. We have adjusted it in line26-27 (“Moreover, the ESCC patients with high STAT3β expression have a complete response to concurrent chemoradiotherapy.”)

Point 12: Line 29 exhibit CCRT sensitivity?

Response 12: In our data, ESCC cell lines such KYSE30 and KYSE150 are more sensitivity to concurrent chemoradiotherapy (CCRT) when STAT3β overexpression. 

Point 13: Line 34 who are not suitable for surgery. Not correct

Response 13: Thank you for the reviewer’s careful work. For the advanced esophageal cancer patients, they are not benefit from surgery lonely. The sentence “Concurrent chemoradiotherapy … who are not suitable for surgery” changed to “Concurrent chemoradiotherapy … patients with advanced esophageal cancer” in line 32-34.

Point 14: Line 37 response for adjuvant chemoradiotherapy (ACRT) marker?

Response 14: Thanks for your hard working. We are sorry for the misunderstanding of this sentence. According to our data, STAT3β could be a marker for indicating the usage of chemoradiotherapy in clinical potentially because of the better response to chemoradiotherapy in ESCC patients with high STAT3β expression. And we also revised this sentence. “STAT3β regulates … for adjuvant chemoradiotherapy (ACRT) marker” have changed to “STAT3β regulates …response to adjuvant chemoradiotherapy (ACRT) potentially” in line 34-36.

Point 15: Line 42 protection factor?

Response 15: Thank you for your careful reading. We have changed “protection” into “protective” in line 41.

Point 16: STAT3α was described in summary but not in abstract

Response 16: We thank the reviewer suggestion. We added STAT3α discussion to the abstract. (line 38-40, “We examined the expression of STAT3α and … STAT3β expression is an independent protective factor (HR=0.424,P = 0.003).)

Point 17: Line 78 pathogen infections?

Response 17: We thank the reviewer for this question. Necroptosis is a feature of cellular clearance of pathogen infection, such as viruses, bacteria, fungi and metazoans [3]. For example, vaccinia virus encodes the caspase inhibitor B13R, which can inhibit caspase function and allow mouse embryonic fibroblasts (MEF) to respond to TNF-α induced necrosis [4]. ZBP1 can senses influenza A virus (IAV) Z-RNAs and Z-form nucleic acids and activation of RIPK3-mediated necroptosis [5,6]. Necroptosis plays a very important role in the pathogenicity of pathogens and related diseases. 

Point 18: Line 93 sensitizes to CCRT?

Response 18: Thank you very much for your reading. We revised and changed “STAT3β sensitizes to CCRT” into “STAT3β increases sensitivity to CCRT (platinum plus radiation therapy)” in line 93-94.

Point 19: The order of tables is wrong.

Response 19: The order of table was corrected and checked carefully. 

Point 20: Almost half patients have stage IV?

Response 20: Thank you for your comment. The ESCC patient’s samples were collected from the Central Hospital of Shantou City. According to the clinical TNM classification (cTNM), ESCC patients were classified into different stage. There were 105 cases of ESCC patients who were received concurrent chemoradiotherapy.  According to the Chinese guidelines for the treatment of esophageal cancer, concurrent chemoradiotherapy should be adopted for patients with locally advanced esophageal cancer who cannot tolerate surgery or who have lost the opportunity of surgery, which is why more stage IV patients were included in this study.

Point 21: Line86 platinum plus ionizing radiation is not the standard regimen for esophageal patients. 

Response 21: Thank you for your question. We agree with the reviewer’s point. What we are trying to say is that concurrent chemoradiotherapy (CCRT) is one of the most promising advanced ESCC treatments. We have changed it. (line 87-89)

Point 22: Figure 2. Do the authors show cytoplasmic STAT3β / nuclear STAT3α or total STAT3β/α in WB as the authors claim the isoform and location of STAT3 may play different role? In vitro study did not separately cytoplasmic and nuclear STAT3. 

Response 22:  We appreciate the reasonable comments from the reviewer. We found that the highly expression of STAT3α and STAT3β in the cytoplasm was significant associated with favorable prognosis of ESCC. And STAT3α highly expressed in the nucleus was associated with poor prognosis. STAT3 is activated by many different signals including cytokines, growth factors and oncogenes and translocate into the nucleus, and initiates transcription. The nuclear localized STAT3α has high transcriptional activity and promotes tumor cell proliferation and survival. STAT3β, a splice variant of STAT3, may play a suppressive effect for it lacks the transactivation domain (TAD). High STAT3β level exerts a tumour suppressive effect. Our lab demonstrated that high STAT3β expression converts the prognostic value of pSTAT3α Y705 from unfavourable to favourable in patients with ESCC [7]. According to the reviewer’s comment, we have separately cytoplasmic and nuclear from KYSE30 and KYSE150 with Vector and STAT3β overexpression. We found STAT3β is more located in the nucleus than STAT3α. The phosphorylation of STAT3α was decreased and there were fewer STAT3α in the nucleus after CCRT in KYSE30 and KYSE150 Vector expressing cells, and STAT3β induced an increase in the expression of pSTAT3α and nuclear retention (Supplementary Figure S3). These data have been added to lines 261 to 263 of the revised manuscript. 

Figure S3. KYSE30 and KYSE150 highly expressing Vector and STAT3β treated with or without cisplatin combined with ionizing radiation, cells were harvested and conducted with nuclear plasma separation. Cyto: cytoplasm; 

Point 23: Figure 3A: STAT3α and STAT3β expressions were noted found by immunoblotting.

Response 23: We appreciate the comments from the reviewer. The band of STAT3β in Figure 3A was not different in vision. We further analyzed the radio of STAT3α and STAT3β by gray value and showed the data in Figure 3A. And in figure3A, we preliminarily screened the expression pattern of two STAT3 isoforms. We found that high-expression of STAT3α but low-expression of STAT3β in these ESCC cell lines in our lab.

Point 24: Figure 3B after 24 hours treatment?
Response 24: We very appreciate for your careful working. We have described in Method 4.8 about this assay. In brief, cells were treated with different concentrations of cisplatin (0, 5, 10, 25, 50 and 100 μM) for 24 hours. After incubated with MTS at 37°C for 2 hours, the results were taken by enzyme-labeled instrument (Multiskan MK3, Thermo Fisher) at 492 nm.

Point 25: Figure 3C decreased pSTAT3?
Response 25: We thank the reviewer for this question. We found pSTAT3 was decreased after cisplatin or ionizing radiation in KYSE150 and KYSE510. To determine this phenomenon, we did the experiment again and used KYSE30 and KYSE150 ESCC cells with different dose of cisplatin combined with ionizing radiation treatment, the phosphorylation of STAT3 was decreased as we can see (Fig. S3). STAT3 activation increases proliferation and survival of tumor cells by upregulating the expression of its downstream genes, such as VEGF and Bcl-2[8]. Our results indicated that downregulating of STAT3 activity (pSTAT3) contributed to ESCC cells growth inhibition and cell death inducing by cisplatin and ionizing radiation.

Legend: KYSE30 and KYSE150 were treated with different dose of DDP for 24 hours, then were ionizing radiation with 5 Gy. Cell were cultured 24 hour and harvested with Laemmli sample buffer. 

Point 26: Figure 1A: did not show nuclear and cytoplasmic STAT3α / STAT3β
Response 26: We thank for your comment. In Figure 1A, B, we show the representative low and high expression of STAT3α (A) and STAT3β (B). We stained STAT3α and STAT3β separately on different slices of the same sample. Thus, we show STAT3α / STAT3β separation. We think the reviewer may want to know why we chose nuclear and cytoplasmic STAT3α / cytoplasmic STAT3β for the combined analysis of prognosis. The expression of nuclear and cytoplasmic STAT3α / STAT3β are evaluated and digitized based upon the intensity of staining according to a Vectra automated multispectral histopathological quantitative analysis system (InForm Version 2.1; PerkinElmer). The expression of nuclear and cytoplasmic are distinguished by machine learning, we found STAT3α distributed in nuclear and cytoplasmic but STAT3β is high expression in cytoplasm and low in nucleus. 

Reviewer 2 Report

This manuscript correlates the high level expression of the transcription factor STAT3β with a favourable prognosis in 105 esophageal squamous cell carcinoma (ESCC) patients. The authors focused their attention on STAT3β signaling pathway and demonstrated that its overexpression enhances sensitivity to chemoradiotherapy through the induction of cell necroptosis. Furthermore, the authors investigated on the molecular mechanisms of this high sensitivity to chemoradiotherapy in different ESCC cell lines.

The manuscript is well-written and the results presented are clear and convincing. There are, however, some issues that require attention.

Major comments

-TE3 cells are more sensitive to DDP treatment while KYSE150 are the most resistant cells. I would like to know why the authors decided to investigate the phosphorylation state of STAT3 on KYSE150 and KYSE510 and not analyzed the effect in TE3 cells. How is the phosphorylation state of STAT3 in TE3 cells? It would be interesting to consider also this parameter.

-In line 175 page 6 the authors wrote: “We found the declined of STAT3 Y705 phosphorylation in KYSE150 and KYSE510 in all treatments”. As shown in Figure 3C, IR and DDT differently modulated STAT3 phosphorylation. Specifically, the KYSE150 are more sensitive than KYSE510 cells to IR and DDP treatments while the DDP treatment in the cell viability is more effective in KYSE510 than KYSE150 cells. Please describe it better in the paragrapher 2.3.

-In Figure 3C the authors check the phosphorylation status of STAT3. The 0.5 µM is the concentration of DDP used, but for how long?

-Figure 4. The authors performed the experiments in KYSE30 and KYE150 cells. Why do the authors select KYSE30 cells? The phosphorylation status of STAT3 was performed in KYSE150 and KYSE510 cells, this figure should be implemented with KYSE510 cells. There are many experiments that are performed switching between cell lines without a uniform designed.

The choice of cells must be maintained.

-Can the increase of expression of STAT3β modulate the phosphorylation of STAT3? In Figure 4, the authors should perform a new experiment like as shown in Figure 3C using the cells transfected with STAT3β.

Minor comments

In lines 196 page 7 and 374 page 13 the authors wrote: “labeled KYSE30-STAT3β and KYSE150-Flag-STAT3β”…“Anti-Flag M2 monoclonal antibody (Sigma-Aldrich”.

Is transfected STAT3β labeled with flag? If yes, I suggest to confirm the presence of transfected STAT3b in the cytosol with confocal microscopy using anti-Flag antibody.

Author Response

Response to Reviewer #2 Comments

This manuscript correlates the high level expression of the transcription factor STAT3β with a favourable prognosis in 105 esophageal squamous cell carcinoma (ESCC) patients. The authors focused their attention on STAT3β signaling pathway and demonstrated that its overexpression enhances sensitivity to chemoradiotherapy through the induction of cell necroptosis. Furthermore, the authors investigated on the molecular mechanisms of this high sensitivity to chemoradiotherapy in different ESCC cell lines.

The manuscript is well-written and the results presented are clear and convincing. There are, however, some issues that require attention.

Major comments

Point 1:TE3 cells are more sensitive to DDP treatment while KYSE150 are the most resistant cells. I would like to know why the authors decided to investigate the phosphorylation state of STAT3 on KYSE150 and KYSE510 and not analyzed the effect in TE3 cells. How is the phosphorylation state of STAT3 in TE3 cells? It would be interesting to consider also this parameter.

Response 1: We thank the reviewer for this suggestion. We apologize for not showing the phosphorylation of STAT3 on TE3. In fact, we investigated the pSTAT3 on five cell lines, and we found that the pSTAT3 changes was the same under DDP or IR treatment. The phosphorylation downregulation of STAT3 should be a common phenomenon in esophageal cancer after treatment with chemoradiotherapy. Our results indicated that downregulating of STAT3 activity (pSTAT3) contributed to ESCC cells growth inhibition and cell death inducing by cisplatin and ionizing radiation. We chose the two that changed significantly. According to reviewer suggestion, we analyzed the phosphorylation state of STAT3 in TE3, data was showed in Fig. 3C. We found the phosphorylation of STAT3 was decreased after treatment with DDP or IR in TE3.

Fig. 3C TE3 was treated with DDP (0.5 μM) or IR (5 Gy) or combination of DDP with IR.

Point 2: In line 175 page 6 the authors wrote: “We found the declined of STAT3 Y705 phosphorylation in KYSE150 and KYSE510 in all treatments”. As shown in Figure 3C, IR and DDP differently modulated STAT3 phosphorylation. Specifically, the KYSE150 are more sensitive than KYSE510 cells to IR and DDP treatments while the DDP treatment in the cell viability is more effective in KYSE510 than KYSE150 cells. Please describe it better in the paragrapher 2.3.

Response 2: Thank you for the reviewer’s comments. We have redescribed in section 2.3, “We found that STAT3α Y705 phosphorylation declined in four cell lines after treatment with DDP or IR, and this was more obvious with the DDP/IR combination (Fig. 3C).” In addition, the ESCC cell lines have different clonogenic ability that can influence the formation of colony. The combination of pSTAT3 and clonogenic assay could be a better indication for discussing chemoradiotherapy sensitivity.

Point 3: In Figure 3C the authors check the phosphorylation status of STAT3. The 0.5 µM is the concentration of DDP used, but for how long?

Response 3: We thank the reviewer for this question. In Figure. 3C, cells were treated with DDP for 24 hours.

Point 4: Figure 4. The authors performed the experiments in KYSE30 and KYE150 cells. Why do the authors select KYSE30 cells? The phosphorylation status of STAT3 was performed in KYSE150 and KYSE510 cells, this figure should be implemented with KYSE510 cells. There are many experiments that are performed switching between cell lines without a uniform designed. The choice of cells must be maintained.

Response 4: Thank you for your careful and valuable comments. In Figure 3E-H, we found KYSE30 was more sensitive to chemoradiotherapy, but KYSE150 was resistant. We wanted to identify STAT3β function from cells with different sensitivities. The pSTAT3 of KYSE30 was added in Fig. 3C, and we also supplement KYSE510 data in Supplementary Figure. S2. We will ensure the integrity of the data and selected cells carefully.

Supplementary Figure S2. The ability of colony formation of high expressed of empty vector and STAT3β in KYSE510 under cisplatin/ X-ray chemoradiotherapy treatment. Left, the representative diagrams of colony formation. Right, statistical histogram of colony formation number. Error bars represent means ± SEM of three independent experiments. #P < 0.05, ##P < 0.01, ###P < 0.001 compared with 0 Gy at the same concentration of DDP; &P < 0.05, &&P < 0.01, &&&P < 0.01 compared with 0 μM DDP at the same dose of IR; *P < 0.05, **P < 0.01, ***P < 0.001 compared with the same Vector treatment.

Point 5: Can the increase of expression of STAT3β modulate the phosphorylation of STAT3? In Figure 4, the authors should perform a new experiment like as shown in Figure 3C using the cells transfected with STAT3β.

We thank the reviewer with a great suggestion. In our previously study, high expression of STAT3β enhanced the phosphorylation of STAT3α Y705 by decreasing the interaction between STAT3α and PTP-MEG2 [7]. According to reviewer suggestion, we perform an experiment to compare the phosphorylation between Vector and STAT3β overexpression under treatment with DDP, IR, DDP+IR. As we can see, the pSTAT3α was decreased after DDP or IR treatment in both Vector and STAT3β groups, but the level of pSTAT3α was higher than STAT3β group (Fig. 4B).

Fig. 4B Immunoblotting analysis. KYSE30 (left) and KYSE150 (right) overexpressing Vector or STAT3β following treatment with vehicle, DDP (0.5 μM), IR (5 Gy), and DDP+IR.

Minor comments

Point 1: In lines 196 page 7 and 374 page 13 the authors wrote: “labeled KYSE30-STAT3β and KYSE150-Flag-STAT3β” “Anti-Flag M2 monoclonal antibody (Sigma-Aldrich”). Is transfected STAT3β labeled with flag? If yes, I suggest to confirm the presence of transfected STAT3β in the cytosol with confocal microscopy using anti-Flag antibody.

We would like to thank the reviewer for the comment to our manuscript. We have KYSE150 expression with Flag-STAT3β. In order to be able to see STAT3α and STAT3β location and expression at the same time, we constructed stably expressed Flag-STAT3α, STAT3β-HA and Flag-STAT3α plus STAT3β-HA in KYSE150 STAT3-KO cells, and detecting Flag and HA tags to determine their location. STAT3α and STAT3β were co-localization and distributed in nucleus and cytoplasm. These data have been added to lines 198 to 202 of the revised manuscript.

B

A

Figure S1 A. KYSE150 STAT3-KO cells transfected with Flag-STAT3α, STAT3β-HA and Flag-STAT3α plus STAT3β-HA, the protein expression were detected by western blotting. B. STAT3α and STAT3β location were detected by Immunofluorescence. Scale bar: 10μM.

Reviewer 3 Report

The study by Zheng et al. investigates the role of the two STAT3 isoforms alpha and beta for the response of esophageal squamous cell carcinoma (ESCC) to concurrent chemoradiotherapy (CCRT). STAT3beta shows a favorable prognosis in 105 ESCC patients with CCRT. In tissue culture, the authors show that overexpression of STAT3beta enhances the sensitivity to CCRT and suggest that TNF signaling and resulting necroptosis are the underlying mechanisms. The study is of interest for the field and is in general well conducted.

While positive prognosis for STAT3beta in ESCC under chemotherapy has been shown earlier, the present study demonstrates this for CCRT and sheds more light on the mechanism.

Major points

  1. Data in Figure 4B and D is not very convincing. For Fig. 4B, only two out of six tests are significant. For Fig. 4D, only two out of nine tests are significant. Results of two additional independent experiments should be shown for B and D to demonstrate that the effect is true. Further, the bars in the bar chart in D should all be arranged in the same order (0 uM, 0.25 uM, 0.5 mM, 1 uM). This will reveal that the number of colonies does not follow an intuitive trend, again arguing for additional experiments that should be shown. I suggest to count the colonies using an alternative approach in addition to ImageJ.
  2. Results, line 309: I don’t think that Fig. 3C showing reduction of pSTAT3 under therapy can be interpreted that STAT3 has an effect on CCRT resistance.

Minor points

  1. A native speaker has to go through the manuscript.
  2. Sentence in the Abstract “In ESCC cells, STAT3β high expression significantly inhibits the ability of colony formation and cell proliferation, thus suggesting that STAT3β enhances sensitivity to CCRT (platinum plus radiation therapy).” I see no support for this conclusion since dormant cells can particularly escape therapy.
  3. Table 2, header: what is meant by “generation dataset”?
  4. Table 2, Note: how have been the thresholds for the scores defined?
  5. Figure 2, the label “All patients with CCRT (n=105)” and “Patients with CCRT (platinum + IR) (n=64)” should be moved to the top (or be on top of both panels, A and B).
  6. The acronym “DMF” should be introduced.
  7. Results, line 231, I suggest to add “TNF signaling” to read “…, 49 TNF signaling genes were up-regulated…”
  8. RIPK1 protein levels for KYSE30 do not look very different between vector and STAT3beta overexpression in the Western blot in Fig. 4G, but this might just be due to the limited dynamic range of the image.
  9. 5, for which cell line(s) have the transcriptomic experiments been done?
  10. 5F, the legend states that 3 independent experiments are shown. The figure shows four experiments.
  11. Discussion, line 272: what is IFI?
  12. Methods, how has the radiation been applied to the tissue culture?
  13. The sample labeling in the headers of the DE genes in the Supp Table spread sheet should match the descriptions in the manuscript or should be explained (what does VT and S3B mean?).
  14. Table S5: how can the rich factor be greater than 1 if it is the number of DEGs in a category divided by the number of all genes in the category. Is there a typo?

Author Response

Response to Reviewer 3 Comments

Major Points

Point 1:  Data in Figure 4B and D is not very convincing. For Fig. 4B, only two out of six tests are significant. For Fig. 4D, only two out of nine tests are significant. Results of two additional independent experiments should be shown for B and D to demonstrate that the effect is true. Further, the bars in the bar chart in D should all be arranged in the same order (0 uM, 0.25 uM, 0.5 mM, 1 uM). This will reveal that the number of colonies does not follow an intuitive trend, again arguing for additional experiments that should be shown. I suggest to count the colonies using an alternative approach in addition to ImageJ.

Response 1: Thank you for your careful and valuable comments. Our previous study indicated that STAT3β decreased the clonogenic ability of esophageal squamous cell carcinoma (ESCC) cells and enhanced the chemosensitivity of ESCC to cisplatin and 5-FU in a STAT3β dose-dependent manner [9]. According to our data of the colony formation assay, we found that STAT3β decreases the formation of cell colony has a chemoradiotherapy dose-dependent manner as well as the control. Compared with the control, STAT3β significantly improved the efficiency of cisplatin and irradiation. As you mentioned, the control group and high expressed STAT3β group as well as the results of these two groups were analyzed. STAT3β could decreased the formation of cell colony and increased the response to chemoradiotherapy. In addition, we further analyzed the other tests and marked the significance. As the figures shows, STAT3β was more responsive to chemoradiotherapy even under a lower concentration of cisplatin and dose of irradiation. In response to the reviewer’s requests, the experiments were repeated and showed the consistent results. Moreover, we are very sorry for making the mistake in Figure 4D arrangement. The data was checked carefully and corrected the figure. According to reviewer suggestion, the colonies were examined and use calculated automatically by Image-Pro Plus. The result is consistent with Image J statistics. Two other independent experiments were shown below.

Point 2: Results, line 309: I don’t think that Fig. 3C showing reduction of pSTAT3 under therapy can be interpreted that STAT3 has an effect on CCRT resistance.

Response 2: Thanks for the reviewer’s thoughtful comments. Indeed, pSTAT3 contributes to chemoradiotherapy resistance. Strong express pSTAT3 was associated with the poor overall survival rate in ESCC patients who underwent chemoradiotherapy [9,10]. Inhibiting the activation of STAT3 by its negative regulator SOCS1 (suppressor of cytokine signaling 1) or its isoform STAT3β, ESCC cells improves the response to chemotherapy and radiotherapy [9,10]. STAT3 expression and IL-6 induced phosphorylation of STAT3 correlated with resistance to chemoradiotherapy in colorectal cancer cells [11]. pSTAT3 in pancreatic adenocarcinoma was also associated with poor clinical outcome [12]. Reducing the level of pSTAT3 sensitized pancreatic cells to gemcitabine and radiotherapy [12]. In our manuscript, the expression of pSTAT3 was reduced under cisplatin and radiation. The data could be one of the evidences to support the STAT3 activity (pSTAT3) contributed to ESCC cells growth inhibition and cell death inducing by cisplatin and ionizing radiation.

Minor Points

Point 1: A native speaker has to go through the manuscript.

Response 1: According to the comments from you, the manuscript had been polished by experts recommended by Cancers.

Point 2: Sentence in the Abstract “In ESCC cells, STAT3β high expression significantly inhibits the ability of colony formation and cell proliferation, thus suggesting that STAT3β enhances sensitivity to CCRT (platinum plus radiation therapy).” I see no support for this conclusion since dormant cells can particularly escape therapy.

Response 2: We are grateful for this excellent comment. STAT3β high expression inhibits the activation of STAT3 and cell proliferation. In our previously study, STAT3β can decreases the cancer stem cell population and sensitizes ESCC cells to chemotherapy [7]. We don’t think it’s a function of inducing cell dormancy. To explain the clinical findings that high expression of STAT3β enhances the sensitivity to CCRT, we did transcriptome analysis and found high expressing STAT3β upregulates TNF and necroptosis signaling pathway compare to control cells after chemoradiotherapy (Fig. 5C). Chemotherapeutic drugs and radiotherapy induced necroptosis have been reported in tumor. It has been found that DDP induced necroptosis through the TNFα-mediated RIPK1/RIPK3/ MLKL pathway. STAT3β enhances ESCC cells necroptosis may be an effective strategy for killing tumor cells even when the cells are dormant.

Point 3: Table 2, header: what is meant by “generation dataset”?

Response 3: Thank you very much to point out the inaccurate description. The header “Univariate and multivariate analysis of STAT3 subtypes associated with overall survival (OS) in generation dataset of ESCC patients with concurrent chemoradiotherapy (n=105).” have been changed to “Univariate and multivariate analysis of STAT3 subtypes associated with overall survival (OS) of esophageal squamous cell carcinoma (ESCC) patients with concurrent chemoradiotherapy (n=105).”

Point 4: Table 2, Note: how have been the thresholds for the scores defined?

Response 4: The thresholds of a protein was defined on the basis of immunohistochemistry scores by using the automatically optimum selection method according to construction of a two-dimensional projection between a biomarker and outcome (X-tile software, Release 3.6.1) [13].

Point 5: Figure 2, the label “All patients with CCRT (n=105)” and “Patients with CCRT (platinum + IR) (n=64)” should be moved to the top (or be on top of both panels, A and B).

Response 5: We redrawn the Figure 2. The label “All patients with CCRT (n=105)” and “Patients with CCRT (platinum + IR) (n=64)” have been moved to the top.

Point 6: The acronym “DMF” should be introduced.

Response 6: Thank you very much. We sincerely apologize to the mistake. DMF is the abbreviation of N, N-Dimethylformamide as a solvent of cisplatin. The acronym “DMF” have been added into the legend of Figure 3D.

Point 7: Results, line 231, I suggest to add “TNF signaling” to read “…, 49 TNF signaling genes were up-regulated…”

Response 7: Thanks for your suggestion. We have added “TNF signaling” in the sentence. “As Fig. 5C shows, 49 genes were upregulated in high expressed STAT3β ESCC cells especially under CCRT treatment” have changed to “As Fig. 5C shows, 49 TNF signaling genes were upregulated in highly expressed STAT3β ESCC cells, especially under CCRT treatment” in line 247-248.

Point 8: RIPK1 protein levels for KYSE30 do not look very different between vector and STAT3beta overexpression in the Western blot in Fig. 4G, but this might just be due to the limited dynamic range of the image.

Response 8: Thank you for your careful and thoughtful comment. The band of RIPK1 in Figure 5G was not different in vision. We further analyzed the intensity of RIPK1 and showed the data in Figure 5H. The data shows that RIPK1 was increased under STAT3β overexpression and chemoradiotherapy. According to the reviewer’s comment, we have repeated the experiment and got the significant results as shown below.

Point 9: for which cell line(s) have the transcriptomic experiments been done?

Response 9: Thank you for your careful read. The transcriptomic experiments were ESCC cell KYSE30. And we have already mentioned in section 4, “Materials and methods”: “KYSE30…performed on BGISEQ-500 system by BGI (Wuhan, China).”.

Point 10: 5F, the legend states that 3 independent experiments are shown. The figure shows four experiments.

Response 10: We very appreciate for your careful working. We divided the experiments into four groups as “Ctrl, IR, DDP and IR+DDP”, each group was the result of three independent experiments.

Point 11: Discussion, line 272: what is IFI?

Response 11: Thank you for your careful work. We are apologized for the carelessness. The IFI is the abbreviation of involved-filed irradiation. We have added the whole name of IFI in the manuscript (line 297).

Point 12: Methods, how has the radiation been applied to the tissue culture?

Response 12: We would like to thank the reviewer’s thoughtful read of our manuscript. We didn’t do tissue culture in this work. The tissue was taken from the patients surgically. As we mentioned in method “Patients samples’, this study was approved by the ethical committee of the Central Hospital of Shantou City (2016-026, 2016.11.4) and the ethical committee of Shantou University Medical College (SUMC-2017-12, 2018.01.01). Written informed consent was obtained from all patients. 105 samples were selected based on pathologic diagnosis and have received current chemoradiotherapy. In addition, we did not culture the tissue but the ESCC cells. Cells were treated with different concentrations of cisplatin (0 μM, 0.25 μM, 0.5 μM, and 1 μM) for 24 h, and different doses of radiation (0 Gy, 1 Gy, 2 Gy, and 4 Gy) using an X-ray Irradiator (RS2000PRO).

Point 13: The sample labeling in the headers of the DE genes in the Supp Table spread sheet should match the descriptions in the manuscript or should be explained (what does VT and S3B mean?).

Response 13: We are very sorry for the mistake. VT and S3B means Vector and STAT3β respectively. We have noted in the supplementary table as Vector and STAT3β.

Point 14: Table S5: how can the rich factor be greater than 1 if it is the number of DEGs in a category divided by the number of all genes in the category. Is there a typo?

Response 14: We are very sorry for the mistake. The rich factor is less than 1. We have checked and corrected the mistake, “1.115” have changed to “0.115”.

References

  1. Chen, S.H.; Chang, J.Y. New Insights into Mechanisms of Cisplatin Resistance: From Tumor Cell to Microenvironment. Int J Mol Sci 2019, 20, doi:10.3390/ijms20174136.
  2. Smith, A.D.; Lu, C.; Payne, D.; Paschall, A.V.; Klement, J.D.; Redd, P.S.; Ibrahim, M.L.; Yang, D.; Han, Q.; Liu, Z., et al. Autocrine IL6-Mediated Activation of the STAT3-DNMT Axis Silences the TNFalpha-RIP1 Necroptosis Pathway to Sustain Survival and Accumulation of Myeloid-Derived Suppressor Cells. Cancer Res 2020, 80, 3145-3156, doi:10.1158/0008-5472.CAN-19-3670.
  3. Baker, M.; Shanmugam, N.; Pham, C.L.L.; Strange, M.; Steain, M.; Sunde, M. RHIM-based protein:protein interactions in microbial defence against programmed cell death by necroptosis. Semin Cell Dev Biol 2020, 99, 86-95, doi:10.1016/j.semcdb.2018.05.004.
  4. Xia, X.; Lei, L.; Wang, S.; Hu, J.; Zhang, G. Necroptosis and its role in infectious diseases. Apoptosis 2020, 25, 169-178, doi:10.1007/s10495-019-01589-x.
  5. Jiao, H.; Wachsmuth, L.; Kumari, S.; Schwarzer, R.; Lin, J.; Eren, R.O.; Fisher, A.; Lane, R.; Young, G.R.; Kassiotis, G., et al. Z-nucleic-acid sensing triggers ZBP1-dependent necroptosis and inflammation. Nature 2020, 580, 391-395, doi:10.1038/s41586-020-2129-8.
  6. Zhang, T.; Yin, C.; Boyd, D.F.; Quarato, G.; Ingram, J.P.; Shubina, M.; Ragan, K.B.; Ishizuka, T.; Crawford, J.C.; Tummers, B., et al. Influenza Virus Z-RNAs Induce ZBP1-Mediated Necroptosis. Cell 2020, 180, 1115-1129 e1113, doi:10.1016/j.cell.2020.02.050.
  7. Zhang, H.F.; Chen, Y.; Wu, C.; Wu, Z.Y.; Tweardy, D.J.; Alshareef, A.; Liao, L.D.; Xue, Y.J.; Wu, J.Y.; Chen, B., et al. The Opposing Function of STAT3 as an Oncoprotein and Tumor Suppressor Is Dictated by the Expression Status of STAT3beta in Esophageal Squamous Cell Carcinoma. Clin Cancer Res 2016, 22, 691-703, doi:10.1158/1078-0432.CCR-15-1253.
  8. Yu, H.; Lee, H.; Herrmann, A.; Buettner, R.; Jove, R. Revisiting STAT3 signalling in cancer: new and unexpected biological functions. Nat Rev Cancer 2014, 14, 736-746, doi:10.1038/nrc3818.
  9. Zhang, H.F.; Chen, Y.; Wu, C.; Wu, Z.Y.; Tweardy, D.J.; Alshareef, A.; Liao, L.D.; Xue, Y.J.; Wu, J.Y.; Chen, B., et al. The Opposing Function of STAT3 as an Oncoprotein and Tumor Suppressor Is Dictated by the Expression Status of STAT3β in Esophageal Squamous Cell Carcinoma. Clinical cancer research : an official journal of the American Association for Cancer Research 2016, 22, 691-703, doi:10.1158/1078-0432.Ccr-15-1253.
  10. Sugase, T.; Takahashi, T.; Serada, S.; Fujimoto, M.; Hiramatsu, K.; Ohkawara, T.; Tanaka, K.; Miyazaki, Y.; Makino, T.; Kurokawa, Y., et al. SOCS1 Gene Therapy Improves Radiosensitivity and Enhances Irradiation-Induced DNA Damage in Esophageal Squamous Cell Carcinoma. Cancer research 2017, 77, 6975-6986, doi:10.1158/0008-5472.Can-17-1525.
  11. Spitzner, M.; Roesler, B.; Bielfeld, C.; Emons, G.; Gaedcke, J.; Wolff, H.A.; Rave-Fränk, M.; Kramer, F.; Beissbarth, T.; Kitz, J., et al. STAT3 inhibition sensitizes colorectal cancer to chemoradiotherapy in vitro and in vivo. International journal of cancer 2014, 134, 997-1007, doi:10.1002/ijc.28429.
  12. Wu, X.; Tang, W.; Marquez, R.T.; Li, K.; Highfill, C.A.; He, F.; Lian, J.; Lin, J.; Fuchs, J.R.; Ji, M., et al. Overcoming chemo/radio-resistance of pancreatic cancer by inhibiting STAT3 signaling. Oncotarget 2016, 7, 11708-11723, doi:10.18632/oncotarget.7336.
  13. Camp, R.L.; Dolled-Filhart, M.; Rimm, D.L. X-tile: a new bio-informatics tool for biomarker assessment and outcome-based cut-point optimization. Clin Cancer Res 2004, 10, 7252-7259, doi:10.1158/1078-0432.CCR-04-0713.
